# ONETWOVLA: A UNIFIED VISION-LANGUAGE-ACTION MODEL WITH ADAPTIVE REASONING

**Fanqi Lin**[1,2,3*]    **Ruiqian Nai**[1,2,3*]    **Yingdong Hu**[1,2,3*]
**Jiacheng You**[1,2]    **Junming Zhao**[1,2]    **Yang Gao**[1,2,3†]

[1]Tsinghua University    [2]Shanghai Qi Zhi Institute    [3]Spirit.AI
*Equal contribution    †Corresponding author    Project page: https://one-two-vla.github.io/

## ABSTRACT

General-purpose robots capable of performing diverse tasks require synergistic reasoning and acting capabilities. However, recent dual-system approaches, which separate high-level reasoning from low-level acting, often suffer from challenges such as limited mutual understanding of capabilities between systems and latency issues. This paper introduces OneTwoVLA, a single unified vision-language-action model that can perform both acting (System One) and reasoning (System Two). Crucially, OneTwoVLA adaptively switches between two modes: explicitly reasoning at critical moments during task execution, and generating actions based on the most recent reasoning at other times. To further unlock OneTwoVLA's reasoning and generalization capabilities, we design a scalable pipeline for synthesizing embodied reasoning-centric vision-language data, used for co-training with robot data. We validate OneTwoVLA's effectiveness through extensive experiments, highlighting its superior performance across four key capabilities: long-horizon task planning, error detection and recovery, natural human-robot interaction, and generalizable visual grounding, enabling the model to perform long-horizon, highly dexterous manipulation tasks such as making hotpot or mixing cocktails. Project page: https://one-two-vla.github.io/.

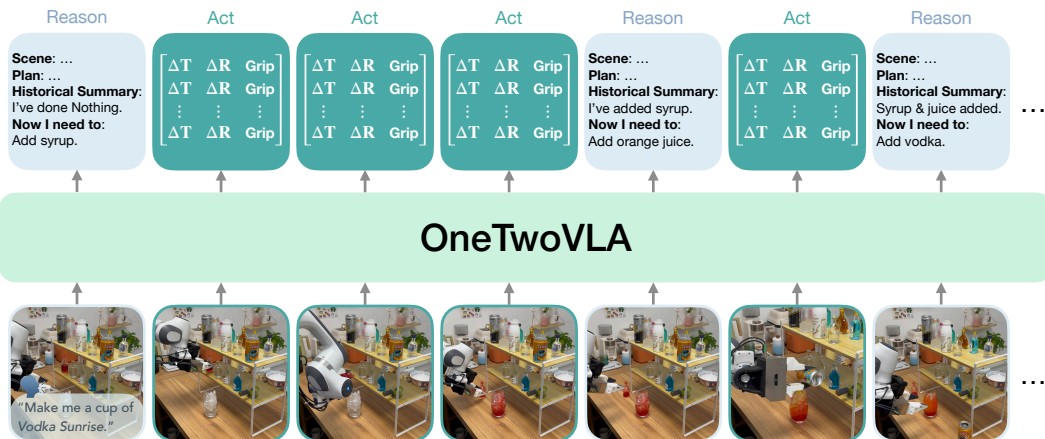

Figure 1: **Overview.** OneTwoVLA is a single unified vision-language-action model capable of both reasoning and acting. Crucially, OneTwoVLA can adaptively reason at critical moments during execution (e.g., upon completing subtasks, detecting errors, or requiring human inputs), while generating actions at other times.

## 1 INTRODUCTION

A distinctive characteristic of human physical intelligence is the ability to both *reason* and *act* (Varela Francisco et al., 1991; Anderson, 2003). Crucially, these processes are not separate but flexibly interleaved, creating a powerful synergy—reasoning guides our actions, while actions provide feedback that informs subsequent reasoning. Consider someone preparing a dish: *reasoning* enables them to develop a comprehensive understanding of the scene and goal (e.g., interpreting the recipe, planning the sequence of steps), while *acting* corresponds to the physical execution (e.g.,

chopping, mixing) that grounds abstract reasoning in the real world. This paper aims to imbue robots with a similar synergistic relationship between reasoning and acting.

Current approaches (Ahn et al., 2022; Hu et al., 2023; Shi et al., 2025; Team et al., 2025; Figure, 2025) often draw inspiration from Kahneman's dual-system framework (Kahneman, 2011). Typically, a System Two, such as internet-pretrained vision-language models (VLMs) (Beyer et al., 2024; Karamcheti et al., 2024), is dedicated to slow high-level reasoning, generating intermediate reasoning contents. Meanwhile, a System One, such as vision-language-action models (VLAs) (Kim et al., 2024; Black et al., 2024; Bjorck et al., 2025), translates these intermediate contents into precise low-level robot actions. However, this explicit decoupling results in both systems lacking mutual awareness of each other's capabilities; System Two may produce intermediate contents that System One cannot execute (Shi et al., 2025). Furthermore, in real-world deployment, issues such as latency may cause System Two to respond belatedly, providing outdated or irrelevant guidance.

We argue that achieving stronger reasoning-acting synergy demands a unified model. Indeed, the recent trend towards unifying capabilities within single models is proving crucial for advancing AI (Yao et al., 2023; OpenAI, 2025), and we believe this approach holds particular promise for robot learning. In light of this, we introduce OneTwoVLA, a single unified vision-language-action model capable of both acting (System One) and reasoning (System Two). Importantly, it adaptively determines when to engage each mode. As shown in Fig. 1, OneTwoVLA triggers natural language reasoning at key steps — like completing a subtask, detecting an error, or requiring human input — producing outputs such as scene descriptions, task plans, historical summaries, and next-step instructions. Otherwise, it generates actions informed by its most recent reasoning outputs. A key advantage of this unified model is its natural support for co-training with vision-language data, significantly enhancing reasoning and generalization. To facilitate this, we develop a scalable pipeline for synthesizing high-quality, embodied reasoning-centric vision-language data.

Our extensive experiments validate OneTwoVLA's effectiveness, demonstrating its ability to integrate diverse capabilities within a single model: **1) Long-horizon task planning:** OneTwoVLA reasons to formulate, track, and dynamically adjust task plans based on execution feedback, significantly outperforming flat VLA (by 30%) and dual-system VLA (by 24%) baselines. Vision-language co-training further enables generalization to novel task instructions (e.g., planning coffee preparation for "Help me stay awake"). **2) Error detection and recovery:** OneTwoVLA detects execution errors in real time, reasons about corrective strategies, and performs agile recovery actions. **3) Natural human-robot interaction:** OneTwoVLA adjusts actions immediately upon human intervention and proactively seeks clarification when faced with ambiguity. **4) Generalizable visual grounding:** OneTwoVLA exhibits superior understanding of spatial relationships, object attributes, and semantic features, even generalizing to objects absent from its robot training data.

## 2 RELATED WORK

**Vision-Language-Action Models.** Initialized from pre-trained vision-language models (VLMs) (Chen et al., 2023; Beyer et al., 2024; Liu et al., 2024a; Wang et al., 2024; Lu et al., 2024), vision-language-action models (VLAs) (Driess et al., 2023; Brohan et al., 2023; Kim et al., 2024; Black et al., 2024; Pertsch et al., 2025; Team et al., 2025; Bjorck et al., 2025; Wen et al., 2025; Huang et al., 2025) have emerged as a promising approach for building general-purpose robots. These VLAs, trained on large robot datasets (Mandlekar et al., 2018; Gupta et al., 2018; Dasari et al., 2019; Cabi et al., 2019; Fang et al., 2020; Brohan et al., 2022; Jang et al., 2022; Walke et al., 2023; O'Neill et al., 2024; Khazatsky et al., 2024; Lin et al., 2024), can handle a wide range of real-world manipulation tasks. However, these VLAs exhibit limited reasoning capabilities (Hu et al., 2023; Shi et al., 2025; Bjorck et al., 2025), showing vulnerability when confronted with long-horizon tasks or complex dynamic environments. Moreover, their generalization performance degrades substantially when facing novel objects or instructions outside the training distribution (Kim et al., 2024; Black et al., 2024). In contrast, our work enhances reasoning and generalization capabilities through a unified model architecture and a co-training framework.

**Reasoning for Robot Control.** Previous works (Stone et al., 2023; Huang et al., 2023; Li et al., 2023a; Belkhale et al., 2024; Liu et al., 2024b; Shi et al., 2024; Zhi et al., 2024; Zhao et al., 2025; Li et al., 2025) demonstrate that high-level reasoning can enhance low-level policy performance in robot control. In particular, many studies (Ahn et al., 2022; Huang et al., 2024; Hu et al., 2023; Shi

**Algorithm 1** Inference Pipeline of OneTwoVLA

**Require:** VLA model $\pi_\theta$, language instruction $\ell$
1:  $t \leftarrow 0$, $I_{\text{ref}}^{1:n} \leftarrow$ initial image, $R \leftarrow$ none
2:  **while** $R \neq$ "Task Finished" **do**
3:      $DT \sim \pi_\theta.decide(\cdot|I_t^{1:n}, I_{\text{ref}}^{1:n}, \ell, R)$
4:      **if** $DT = $ [BOR] **then**
5:          $\hat{R} \sim \pi_\theta.reason(\cdot|I_t^{1:n}, I_{\text{ref}}^{1:n}, \ell, R)$
6:          $R \leftarrow \hat{R}$, $I^{\text{ref}} \leftarrow I_t$
7:      **else if** $DT = $ [BOA] **then**
8:          $A_t \sim \pi_\theta.act(\cdot|I_t^{1:n}, I_{\text{ref}}^{1:n}, \ell, R, s_t)$
9:          Execute $A_t$
10:      **end if**
11:      $t \leftarrow t + 1$
12: **end while**

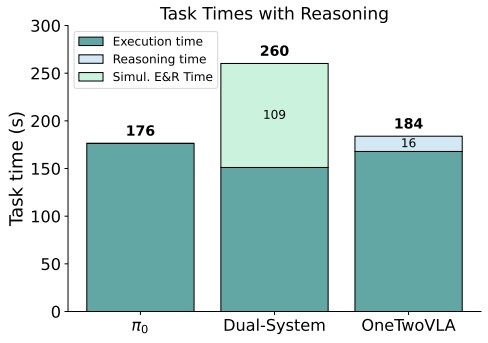

Figure 2: **Task completion times on `Tomato-Egg`.** For experimental settings, see Sec. 4.1.

et al., 2025; Team et al., 2025; Bjorck et al., 2025; Figure, 2025) explore dual-system frameworks, where a foundation model (e.g., a VLM) serves as System Two to perform high-level reasoning, while a low-level policy operates as System One to generate actions based on reasoning outputs. While this dual-system framework proves effective for accomplishing long-horizon manipulation tasks, it inherently suffers from limitations such as the two systems lacking mutual awareness of each other's capabilities (Shi et al., 2025) as well as latency issues with System Two. Recently, $\pi_{0.5}$ (Intelligence et al., 2025) employs a single model to predict a subtask before each action, but this reasoning is simple and information-limited. If this inflexible paradigm generates extensive reasoning at every step, it significantly impacts inference efficiency (Zawalski et al., 2024). To mitigate this, ECoT-Lite (Chen et al., 2025) avoids producing reasoning during test time, but this leads to degraded performance and prevents effective human-robot interaction. To address these limitations, we propose a unified model capable of adaptively deciding when to reason versus when to act, allowing for both informative reasoning and efficient execution. For related work on co-training for robot learning, please refer to the Appendix B.

## 3 METHOD

In this section, we first introduce the framework of OneTwoVLA in Sec. 3.1, including its formulation, adaptive inference, and model instantiation. We then describe how we curate robot data to enable synergistic reasoning and acting in Sec. 3.2. Finally, we present our scalable pipeline for synthesizing vision-language data enriched with embodied reasoning in Sec. 3.3.

### 3.1 FRAMEWORK OF ONETWOVLA

**Problem Formulation.** The central problem investigated in this work is how to develop a robotic control policy $\pi_\theta$ capable of both reasoning and acting, with the critical ability to autonomously decide at each timestep $t$ whether to reason or act. Formally, the policy operates in two modes. When in reasoning mode, the policy takes as input the current image observations from multiple cameras $I_t^1, \ldots, I_t^n$ (denoted as $I_t^{1:n}$, where $n$ is the number of cameras), the reference images from the latest reasoning timestep $I_{\text{ref}}^1, \ldots, I_{\text{ref}}^n$ (denoted as $I_{\text{ref}}^{1:n}$, which introduces observation histories to prevent ambiguous states), the language instruction $\ell$, and the latest reasoning content $R$. The policy performs reasoning in the form of textual output, generating updated reasoning content $\hat{R} \sim \pi_\theta(\cdot|I_t^{1:n}, I_{\text{ref}}^{1:n}, \ell, R)$. Sec. 3.2 provides further details on the specific content of this reasoning process. In acting mode, the policy $\pi$ additionally incorporates the robot's proprioceptive state $s_t$ and generates an action chunk $A_t$ based on the latest reasoning content: $A_t \sim \pi_\theta(\cdot|I_t^{1:n}, I_{\text{ref}}^{1:n}, \ell, R, s_t)$.

**Adaptive Inference of OneTwoVLA.** In Algorithm 1, we present the detailed process of how OneTwoVLA autonomously decides whether to reason or act. We introduce two special decision tokens ($DT$): *beginning of reasoning* ([BOR]) and *beginning of action* ([BOA]). Given the prefix (comprising image observations $I_t^{1:n}$, reference images $I_{\text{ref}}^{1:n}$, instruction $\ell$, and the latest reasoning content $R$), the model first predicts either [BOR] or [BOA]. When [BOR] is predicted, the model enters reasoning mode and generates textual reasoning content $R$ until producing an *end of sentence* ([EOS]) token. Conversely, when [BOA] is predicted, the model enters acting mode and directly generates the action chunk $A_t$. This adaptive framework yields high inference efficiency: during task execution, the model operates primarily in acting mode, invoking reasoning only at a few crit-

ical steps, which adds only minimal overhead. As shown in Fig. 2, for completing a long-horizon task, OneTwoVLA achieves total times that match those of a flat VLA without language reasoning ($\pi_0$ (Black et al., 2024)). In contrast, a Dual-System approach that "always reasons" (e.g., Hi Robot (Shi et al., 2025), ViLa (Hu et al., 2023)) incurs significant latency due to extensive reasoning. Moreover, our framework inherently supports error recovery and human-robot interaction: when the policy detects an error (e.g., failing to grasp an object), it autonomously enters reasoning mode to determine a corrective strategy and execute agile recovery actions. When human interaction occurs, any interaction text will be consistently added to the language instruction $\ell$ in subsequent steps.

**Model Instantiation.** OneTwoVLA is designed to be general, allowing most existing VLAs to be integrated with minimal modifications. For a specific instance, we employ $\pi_0$ (Black et al., 2024) as the base VLA, which demonstrates strong performance across various tasks. The vision-language model of $\pi_0$ auto-regressively generates textual reasoning during inference and is supervised via a cross-entropy loss during training. To model complex continuous action distributions, we inherit the action expert architecture from $\pi_0$ and train it using a flow matching loss (Lipman et al., 2022; Liu, 2022). OneTwoVLA's inference flow is detailed in Fig. 3. See Appendix F.2 for more training details.

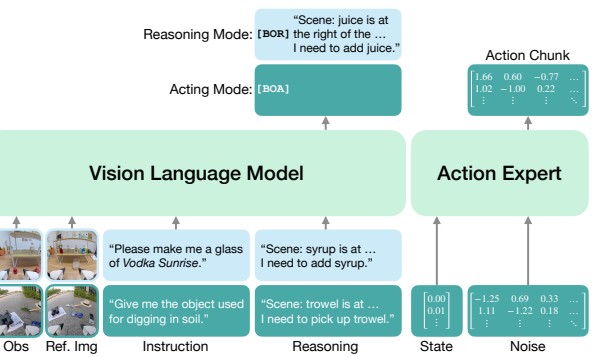

Figure 3: **Inference flow** of OneTwoVLA in two modes.

## 3.2 Curating Robot Data with Embodied Reasoning

Most existing robotic manipulation datasets consist primarily of observation-action pairs and lack associated reasoning information. To address this gap, we introduce a novel robot data format. For a given task, we first collect demonstration trajectories provided by human experts. Subsequently, each trajectory is segmented into a sequence of intervals. There are two types of intervals: *reasoning intervals*, which capture key steps requiring model reasoning (e.g., upon completing subtasks, detecting errors, or when human interaction is required), which we further annotate with textual reasoning content; and *acting intervals*, in which the model primarily learns to predict actions based on observations and the latest reasoning content. During training, we supervise the decision tokens according to the interval type and the freshness of reasoning content $R$. In reasoning intervals, the ground-truth decision token is [BOR] if the current reasoning $R$ is stale (i.e., needs updating); once $R$ has been updated, the ground truth becomes [BOA]. In acting intervals, the model always learns to predict [BOA]. See Appendix F.1 for more details.

Next, we elaborate on the embodied reasoning content. As shown in Fig. 4 left, it consists of four components: 1) a detailed *scene description*, primarily focusing on the locations of task-relevant objects; 2) a *high-level plan* that outlines the sequential steps to accomplish the task; 3) a concise *historical summary* to keep the model informed about the task's progress; and 4) the immediate *next step* that the robot needs to execute. This comprehensive reasoning content encourages the model to understand the visual world, learn high-level planning, and track task progress. Furthermore, to equip the policy with error detection and recovery capabilities, we specifically collect and label robot data focused on recovery from failure states. To enable natural human-robot interaction, we annotate certain intervals of the demonstrations with interaction context (e.g., the robot's question and the human's answer shown in Fig. 4 left).

We design a two-stage *fully automated pipeline* for labeling reasoning intervals and generating their reasoning content. In the first stage, *interval annotation*, we predefine a high-level plan with $K$ ordered subtasks $P = (p_1, \ldots, p_K)$. From each demonstration, we uniformly subsample $N = 32$ frames $S = \{I_{t_n}\}_{n=1}^N$ and prompt Gemini 2.5 to identify reasoning intervals immediately after each subtask completion, yielding $K + 1$ intervals $\mathcal{R} = \{(r_j^s, r_j^e)\}_{j=0}^K$. In the second stage, *reasoning content generation*, for each interval $(r_j^s, r_j^e) \in \mathcal{R}$ we take its midpoint frame and denote it as $\hat{I}_j$. We then construct four reasoning fields: 1) a scene description $D_j$ generated by Gemini from $\hat{I}_j$; 2) the high-level plan $P$; 3) a historical summary $H_j = (p_1, \ldots, p_j)$; and 4) the next step $X_j = p_{j+1}$.

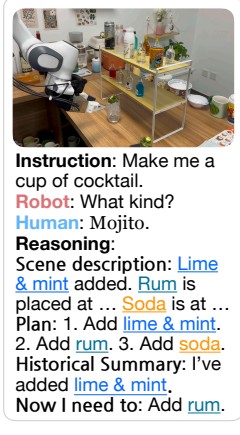

**Instruction**: Make me a cup of cocktail.
**Robot**: What kind?
**Human**: Mojito.
**Reasoning**:
Scene description: Lime & mint added. Rum is placed at … Soda is at …
Plan: 1. Add lime & mint. 2. Add rum. 3. Add soda.
Historical Summary: I've added lime & mint. Now I need to: Add rum.

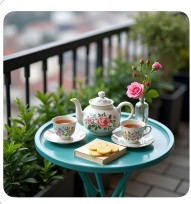

**Instructions:**
Spatial: Get the item between teacups.
Semantic: Get the item for pouring tea.
Attribute: Hand me the object with spout.
**Reasoning**: Pick up teapot between cups.

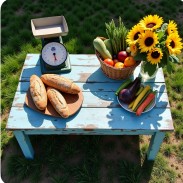

**Instructions:**
Spatial: Get the item in front of the scale.
Semantic: Pass me the French food.
Attribute: I want that crusty baked good.
**Reasoning**: Pick up bread on round board.

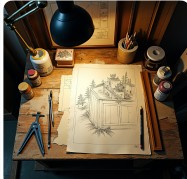

**Instruction:**
Clean up table.
**Reasoning**:
Plan: 1. Put compass in drawer. 2. Put black pencil in holder. 3. Put ruler in drawer. 4. Fold drawing. 5. Put folded drawing in drawer.

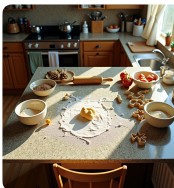

**Instruction:**
Bake cookies.
**Reasoning**:
Plan: 1. Roll out white dough. 2. Cut shapes from dough. 3. Bake cookies. 4. Take out baked cookies from oven.

Figure 4: **Left.** Example of *robot data* with reasoning content. The reasoning content comprises a scene description, a high-level plan, a historical summary, and the next-step instruction. Interaction texts (e.g., the robot question and the human answer) are appended after the instruction. **Right.** Examples of synthetic embodied reasoning-centric *vision-language data*. The top two examples illustrate visual grounding tasks, while the bottom two demonstrate long-horizon tasks. More examples are provided in Appendix E.

The tuple $(D_j, P, H_j, X_j)$ serves as the reasoning content for interval $j$. This automated pipeline produces high-quality annotations. For example, in the `Tomato-Egg` task, 81.5% of intervals are judged correct by human annotators, and 83.3% of scene descriptions are deemed reasonable. Additional prompts and extensions to error recovery are provided in Appendix D.

## 3.3 SCALABLE SYNTHESIS OF VISION-LANGUAGE DATA WITH EMBODIED REASONING

The carefully curated robot data described in Sec. 3.2 allows the model to directly learn the desired task, but its size scales linearly with the costly human effort, making large dataset creation impractical. To endow our model with stronger generalization and the ability to cope with highly varied scenarios, we leverage off-the-shelf foundation models and design a fully scalable pipeline that synthesizes vision-language data enriched with embodied reasoning. This pipeline consists of three steps: 1) We prompt Gemini 2.5 Pro to generate diverse textual descriptions of tabletop layouts featuring common household items; 2) Based on these textual descriptions, we employ the text-to-image generation model FLUX.1-dev (Labs, 2024) to synthesize high-quality images depicting the tabletop layouts. We further augment the synthetic images by randomly applying fisheye distortion or compositing a robot gripper with adaptive brightness, making the visuals more closely resemble real robot observations; 3) Finally, we utilize Gemini again to generate task instructions and corresponding reasoning contents for each synthesized image. Through this pipeline, we automatically generated 16,000 data samples, with examples shown in Fig. 4 right.

The generated task instructions fall into two categories: 1) Visual grounding tasks (Shridhar & Hsu, 2018; Bhat et al., 2024; Kim et al., 2023), where the instruction implicitly refers to an object in the image through spatial relationships, attributes, or semantic features. The accompanying reasoning must reveal the object's explicit name and, optionally, its location; 2) Long-horizon tasks, where the instruction describes an extended, multi-step objective. The reasoning must supply a high-level, step-by-step plan for completing the task. For part of the dataset, we additionally instruct Gemini to incorporate elements of human–robot interaction into the plan. A detailed quality analysis of the synthetic data, along with additional examples, is provided in Appendix E.

## 4 EXPERIMENTS

In this section, we evaluate OneTwoVLA through extensive real-world experiments, demonstrating its superior performance in versatile capabilities: long-horizon task planning (Sec. 4.1), error detection and recovery (Sec. 4.2), natural human-robot interaction (Sec. 4.3), and visual grounding (Sec. 4.4). Additionally, we show that co-training with our synthetic vision-language data yields generalizable behaviors and open-world visual grounding capabilities on unseen scenarios and tasks.

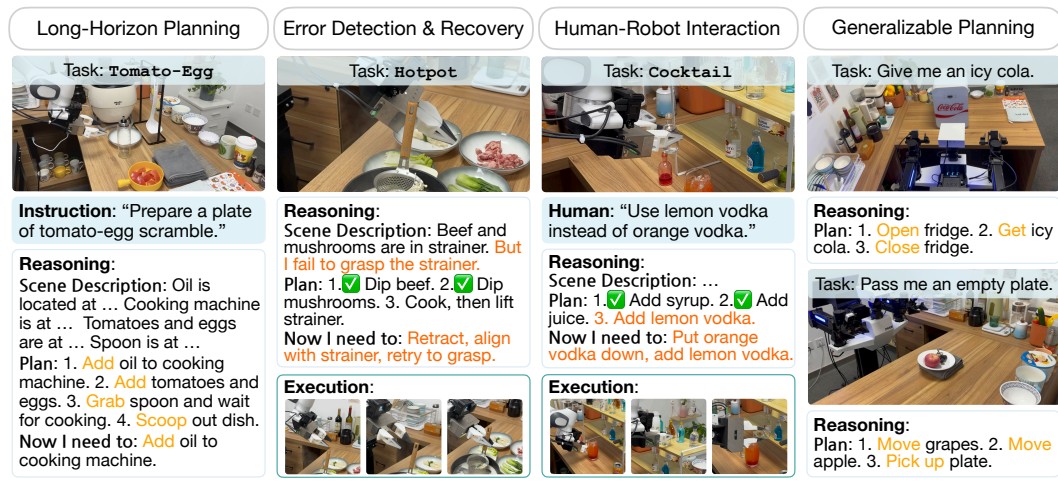

Figure 5: **Task illustrations and reasoning examples.** In the three leftmost columns, we present three challenging, long-horizon manipulation tasks. Completing these tasks requires not only planning abilities, but also error detection and recovery capabilities, as well as the the ability to interact naturally with humans. In the rightmost column, we demonstrate two tasks drawn from our experiments on generalizable planning. For every task, we include a sample of the model's reasoning content. See Appendix D for additional reasoning examples.

## 4.1 LONG-HORIZON TASK PLANNING

**Hardware.** We utilize two robot platforms. The primary platform consists of a single 7-DoF Franka arm equipped with a parallel jaw gripper. A wrist-mounted GoPro camera with fisheye lens provides wide field-of-view observations. Most of our experiments are conducted using this setup. Additionally, we employ a dual-arm platform featuring two 6-DoF ARX arms with three cameras (two wrist and one base), primarily for generalizable planning experiments. See Appendix H for further details.

**Long-horizon Tasks.** We design three challenging long-horizon tasks (shown in Fig. 5), each requiring the robot to understand the scene, plan accordingly, accurately track task progress, and generate precise actions throughout execution. We briefly describe these tasks here, with more details provided in Appendix C.1: 1) `Tomato-Egg`: The robot pours oil followed by tomato and egg liquid into a cooking machine. Once cooking completes, it uses a spoon to scoop the scramble onto a plate—a contact-rich action demanding fine precision. 2) `Hotpot`: Four plates containing different food items are placed on the table, and their relative ordering is randomized. The robot must sequentially dip beef and one vegetable type, precisely place them into a strainer, and finally lift the strainer. 3) `Cocktail`: The robot mixes one of three cocktails (`Mojito`, `Mountain Fuji`, or `Vodka Sunrise`), each requiring 3-4 steps of ingredient pouring. The robot must distinguish between nearly ten visually similar ingredients and pour accurately. For all tasks, the initial placement of all manipulated objects is randomized within a $10 \times 10$ cm$^2$ area.

**Baselines.** We compare OneTwoVLA against two baselines: 1) a state-of-the-art VLA model $\pi_0$ (Black et al., 2024), which does not perform reasoning. To ensure fair comparison, we fine-tune $\pi_0$ on the same dataset used for training OneTwoVLA; and 2) a dual-system approach inspired by Hi Robot (Shi et al., 2025), in which Gemini 2.5 Pro serves as the high-level System Two that decomposes complex instructions into sequences of atomic commands (i.e., the *next step* field in OneTwoVLA's reasoning content). In practice, we invoke System Two at fixed intervals (i.e., effectively "always reasoning"). For System One, we annotate our dataset with atomic commands and fine-tune $\pi_0$ to execute the commands produced by System Two.

**Experimental Results.** As shown in Fig. 6 left, OneTwoVLA achieves an average success rate of 87% across the three challenging tasks, outperforming $\pi_0$ by 30% and the dual-system approach by 24%. OneTwoVLA consistently generates correct plans, accurately tracks task progress, and outputs precise actions. In contrast, lacking explicit reasoning and historical context, $\pi_0$ sometimes loses track of its current step — such as staying stuck at the initial position when preparing `Mojito` or repeatedly picking up beef in the `Hotpot` task. We also observe that explicit reasoning facilitates more fine-grained action learning; $\pi_0$ sometimes struggles to grasp ingredients precisely in `Hotpot` task or scoops too lightly in the `Tomato-Egg` task, whereas OneTwoVLA performs these delicate actions accurately. Regarding the dual-system approach, we found limitations arising

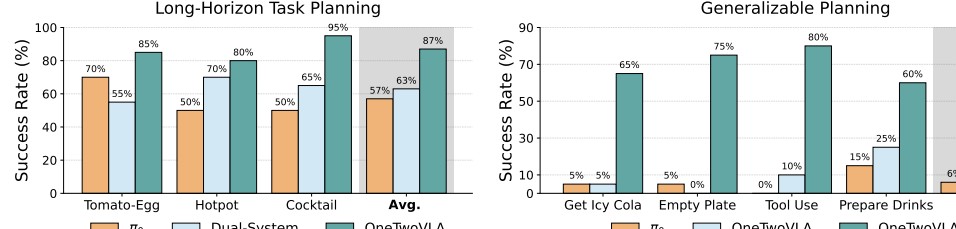

Figure 6: **Left: evaluation results on long-horizon tasks.** OneTwoVLA excels in long-horizon task planning compared to baselines. **Right: evaluation results on generalizable planning tasks.** By co-training with synthetic vision–language data, OneTwoVLA-VL further enhances its generalization to novel tasks. In both figures, each method is evaluated over 20 trials per task.

from the lack of mutual awareness between the two systems' capabilities. System Two occasionally outputs atomic commands that are infeasible for System One to execute (e.g., instructing to add green onion in the `Tomato-Egg` task when none is present). Additionally, the significant inference latency of Gemini 2.5 Pro may prevent System Two from promptly updating its reasoning content, causing System One to encounter out-of-distribution states during execution.

**Generalizable Planning.** We investigate how co-training with vision–language (VL) data can improve OneTwoVLA's ability to generalize in task planning. Specifically, we collect additional demonstration data for various atomic skills (e.g., pick, place, open, etc.) across two robot platforms. We then co-train OneTwoVLA on these robot data together with the with 16,000 VL samples synthesized by the pipeline described in Sec. 3.3. We denote this variant as **OneTwoVLA-VL**, while the model trained exclusively on robot data is denoted as OneTwoVLA. During evaluation, the policy receives instructions that *never appear* in the robot data (e.g., Fig. 5, last column). We test on four challenging tasks that demand deep commonsense reasoning (see Appendix C.2 for details).

As shown in Fig. 6 right, OneTwoVLA-VL exhibits strong generalization, transferring knowledge from VL data to robot control. The model proactively searches for non-visible cola by opening the refrigerator in `Get Icy Cola`, handles complex spatial relationships and occlusion by first removing fruits before retrieving the plate in `Empty Plate`, recognizes the need for a tool and uses a nearby stick to sweep distant objects within reach in `Tool Use`. Furthermore, it exhibits sophisticated scene-aware human intent understanding in `Prepare Drinks`, preparing coffee for "Help me stay awake", kale juice for "I want something healthy". In contrast, $\pi_0$ executes random atomic skills when faced with such novel tasks, and OneTwoVLA without VL co-training produces entirely incorrect plans—both exhibiting only minimal generalizable planning abilities.

## 4.2 ERROR DETECTION AND RECOVERY

Recovering from mistakes is a critical capability for general-purpose robots. OneTwoVLA can detect errors in real-time, rapidly reason about recovery strategies, and subsequently generate corrective actions learned from collected robot recovery data. Table 1 presents a quantitative comparison of error recovery performance on two tasks: `Hotpot` and `Tomato-Egg`. In

|  | Hotpot | Tomato-Egg | Total |
|---|---|---|---|
| OneTwoVLA | 5 / 6 | 3 / 4 | **8 / 10 (80.0%)** |
| $\pi_0$ | 3 / 7 | 5 / 7 | 8 / 14 (57.1%) |
| Dual-System | 4 / 5 | 3 / 7 | 7 / 12 (58.3%) |

Table 1: **Error Detection and Recovery Results.** Values are reported as # successful recoveries / # error occurrences.

the `Hotpot` task, the robot occasionally fails to grasp the strainer due to misalignment. We therefore collect 200 demonstrations containing recovery actions (600 demonstrations in total). After retraining, OneTwoVLA reasons to retract, reposition to align with the strainer and try grasping again, subsequently succeeding in lifting it up. In contrast, $\pi_0$ frequently ignores errors and continues to lift the gripper despite not having grasped the strainer. In the `Tomato-Egg` task, sometimes the oil bottle slips from the gripper while pouring; we collect 100 recovery demonstrations (200 in total). OneTwoVLA recognizes the error, reasons to adjust its grasp for increased firmness and retry the action. However, the dual-system approach fails to respond promptly due to latency issues. System Two only alerts that the oil bottle is not grasped after the robot has already reached the pouring pose, by which time recovery is hard because the robot has entered an out-of-distribution state.

### 4.3 NATURAL HUMAN-ROBOT INTERACTION

Deploying robots in human-centric scenarios requires natural interaction with people. We conduct 10 human-robot interactions for OneTwoVLA and the Dual-System baseline ($\pi_0$ is omitted as it cannot generate language outputs) on the `Hotpot` and `Cocktail` tasks, with subtask-level interaction results shown in Table 2.

Due to its adaptive nature and explicit reasoning process, OneTwoVLA is able to engage with humans in a natural way — seamlessly handling human interventions and proactively seek clarification when faced with ambiguities. For example, in the `Hotpot` task, when a human interrupts by requesting, "Could you also dip another vegetable for me?" OneTwoVLA immediately responds by clarifying,

|  | Hotpot | Cocktail | Total |
|---|---|---|---|
| OneTwoVLA | 10 / 10 | 10 / 10 | **20 / 20 (100%)** |
| Dual-System | 8 / 10 | 5 / 10 | 13 / 20 (65%) |

Table 2: **Human-Robot Interaction Results.** Each entry is reported as # successes / # interactions.

"Sure! Would you like green bok choy, enoki mushrooms, or cabbage?" In the `Cocktail` task, when the robot is preparing a `Vodka Sunrise` and the human interrupts with, "I don't want orange vodka, I want lemon-flavored one," OneTwoVLA immediately reasons that it needs to put down the orange vodka, retrieve the lemon vodka, and generate action sequences that align with the human's intent. In contrast, the dual-system approach frequently loses context during interaction and struggles to maintain a coherent reasoning process, merely picking up the lemon vodka without continuing to prepare the cocktail in the example above.

**Generalizable Human-Robot Interaction.** We further evaluate generalization on the `Hotpot` and `Cocktail` tasks by testing 20 novel interaction scenarios that *never appear* in the robot training data. With vision–language data co-training, OneTwoVLA-VL achieves a 72.5% success rate. In the `Hotpot` task, OneTwoVLA-VL demonstrates proactive reasoning by asking "Which plate of meat would you like me to cook?" when given the instruction "cook meat" with two meat plates present. It also effectively handles unseen dynamic interruptions — when picking up enoki mushrooms, if interrupted with, "I don't want enoki mushrooms, please cook some green bok choy instead," it immediately switches to executing the new instruction. In contrast, OneTwoVLA without VL data co-training fail to interpret such unseen interaction commands, exhibiting poor generalization.

### 4.4 ENHANCED VISUAL GROUNDING

Grounding objects in language instructions to the visual world is a prerequisite for robots to accomplish more complex tasks. We categorize visual grounding into three key aspects (Shridhar & Hsu, 2018; Bhat et al., 2024; Kim et al., 2023; Shridhar et al., 2020): spatial relationships, object attributes, and semantic features. To validate OneTwoVLA's effectiveness in these aspects, we design experiments where instruction following requires non-trivial object grounding capabilities. Furthermore, to demonstrate the impact of our synthetic vision-language data, we conduct experiments in open-world settings where diverse items and environments pose additional challenges. The specific experimental settings are described below (shown in Fig. 7):

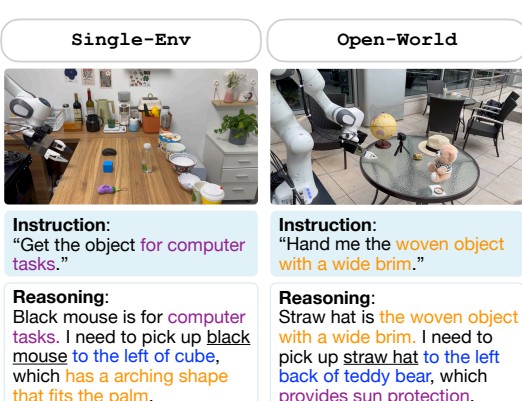

Figure 7: **Illustrations of visual grounding tasks.** In the `Single-Env` setting, we provide task instructions that require understanding of spatial relationships, object attributes, or semantic features. In the `Open-World` setting, we further evaluate the model's *generalizable* visual grounding capabilities.

1) `Single-Env`: Four objects are randomly arranged on a tabletop in a single environment. We collect 50 picking-up demonstrations for each object using the UMI (Chi et al., 2024) device, totaling 200 demonstrations. For testing, we perform 40 trials per method in the same environment using the same four objects. 2) `Open-World`: We collect demonstrations in 16 diverse in-the-wild environments, totaling 933 valid demonstrations using the UMI device. Each demonstration involves moving the gripper to a randomly selected object within the scene, collectively including 180 distinct household items. For testing, we evaluate each method across 8 unseen environments, testing

5 times per environment, each time randomly selecting one from 20 objects: 5 objects seen in robot data, 10 objects unseen in robot data but present in synthetic vision-language data, and 5 objects unseen in either dataset. In both settings, training and test instructions refer to target objects using their names or through spatial relationships, attributes, or semantic features. Our annotated reasoning explicitly identifies the target object's name and includes additional information about it. We compare three methods: $\pi_0$, OneTwoVLA, and OneTwoVLA-VL. Both $\pi_0$ and OneTwoVLA are trained exclusively on robot data, whereas OneTwoVLA-VL is additionally co-trained with 16,000 synthetic vision–language samples. Further details are provided in Appendix C.3.

**Explicit reasoning facilitates visual grounding.** In the `Single-Env` setting, as shown in Table 3, OneTwoVLA achieves a success rate of 78%, significantly outperforming $\pi_0$, which has a success rate of only 5%. In most cases, OneTwoVLA accurately interprets spatial relationships, object attributes, and semantic features described in the instructions, reasons about the correct object, and then successfully picks up the target object. In stark contrast, $\pi_0$ consistently fails to comprehend the instructions, even when the target object is explicitly named. $\pi_0$ typ-

|  | Single-Env | Open-World |
|---|---|---|
| $\pi_0$ | 5% | 3% |
| OneTwoVLA | 78% | 8% |
| OneTwoVLA-VL | 88% | 73% |

Table 3: **Evaluation results on visual grounding tasks.** OneTwoVLA exhibits strong visual grounding capabilities, and co-training with VL data further enhances its generalization.

ically extends the gripper forward aimlessly or randomly picks up the closest object. This clear performance gap demonstrates that explicitly learning to reason helps the model truly understand the visual world rather than attempting to find shortcuts to overfit actions. Moreover, we find that the reasoning content also aids the model in fitting actions, as evidenced by $\pi_0$'s action mean squared error (MSE) on the validation set being 62% higher than OneTwoVLA's.

**Reasoning-centric vision-language data enables generalizable visual grounding.** In the `Open-World` setting, OneTwoVLA-VL achieves a 73% success rate, significantly outperforming both OneTwoVLA and $\pi_0$. In most cases, OneTwoVLA-VL can correctly handle objects unseen in the robot data but present in vision-language (VL) data, effectively transferring commonsense knowledge from VL data to the robot policy. Remarkably, OneTwoVLA-VL generalizes even to novel objects that appear in neither the robot nor VL training data (e.g., Sprite, GoPro). We attribute this exceptional generalization capability to VL data co-training, which better activates web knowledge already encoded in the pretrained vision-language model. In contrast, OneTwoVLA and $\pi_0$ frequently exhibit aimless reaching behaviors — even for objects present in the training data — indicating that they merely overfit to action training data without developing genuine understanding of the visual environment in this complex and diverse setting.

## 5   CONCLUSION, LIMITATIONS AND FUTURE WORK

In this paper, we present OneTwoVLA, a single unified model capable of both reasoning and acting, and adaptively switching between these two modes. This synergy is enabled by our meticulously designed framework and reasoning-enriched robot data curation. Moreover, we propose a scalable pipeline for synthesizing embodied reasoning-centric vision-language data to further enhance the model's reasoning and generalization capabilities. Extensive experiments demonstrate OneTwoVLA's superior performance across four key abilities: long-horizon task planning, error detection and recovery, natural human-robot interaction, and generalizable visual grounding.

There are several limitations that future work can address. First, OneTwoVLA relies on hand-selected heuristics to determine which steps require reasoning. Analogous to supervised fine-tuning (SFT) in LLMs, this strategy aligns the policy with a set of candidate reasoning steps and content; however, achieving more optimal reasoning likely requires reinforcement learning (RL). Future work could investigate RL-based training to further strengthen the reasoning capabilities of VLA models. Second, although our adaptive framework allows the model to reason only at a few critical steps during task execution, the robot still needs to pause for two to three seconds while reasoning occurs. Future research could explore the design of asynchronous architectures, enabling simultaneous reasoning and action generation. Third, we have not optimized action inference efficiency. As our single unified model scales to larger parameter sizes, the time cost of action inference may become a bottleneck. We anticipate that incorporating advanced inference techniques from the LLM literature could help accelerate action inference in future work. Finally, due to resource constraints, we only investigate the effect of high-quality synthetic vision-language data on VLA reasoning capabilities. Future work could explore the impact of vision-language data from various sources.

ACKNOWLEDGMENTS

This research was conducted with the support of the Shanghai Qi Zhi Institute & Spirit AI Innovation Program and the Tsinghua University Dushi Program. Funding and support for this work were also provided by the Tsinghua University - Keystone Electrical (Zhejiang) Co.,Ltd Joint Research Center for Embodied Multimodal Artificial Intelligence (JCEMAI).

We thank Tong Zhang, Chuan Wen, Weirui Ye, Weijun Dong, Shengjie Wang, Chengbo Yuan, Boyuan Zheng, Haoxu Huang, Yihang Hu, and Yuyang Liu for valuable discussions. We also thank the ARX team for prompt assistance with robot hardware and the Xiongan AI Institute for their support.

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

## APPENDIX

Please visit our website to view robot rollout videos: https://one-two-vla.github.io/.

## A    LARGE LANGUAGE MODELS USAGE STATEMENT

We used large language models only for light grammar checking and language polishing. All content was written by the authors.

## B    MORE RELATED WORK

**Co-training for Robot Learning.** Co-training with data from diverse sources has been shown to benefit robot learning (Vuong et al., 2023; Driess et al., 2023; Li et al., 2023b; Nasiriany et al., 2024; Hejna et al., 2024; Yang et al., 2024; Doshi et al., 2024; Yuan et al., 2024; Maddukuri et al., 2025). In particular, several prior works (Brohan et al., 2023; Mu et al., 2023; Zhu et al., 2025; Zhou et al., 2025)explore co-training robot policies with action-free vision-language data alongside robot data, demonstrating improvements in policy generalization. However, these methods (Brohan et al., 2023; Mu et al., 2023; Zhou et al., 2025) typically either rely on existing vision-language datasets, which suffer from limited quality due to their significant domain gap from robot application scenarios; or manually collect vision-language datasets (Zhu et al., 2025), which are inherently limited in size and difficult to scale up. To address these limitations, we propose a scalable pipeline for synthesizing vision-language data rich in embodied reasoning. Our pipeline ensures both high quality and scalability, significantly enhancing policy's reasoning and generalization capabilities.

## C TASKS AND EVALUATIONS

In this section, we provide a detailed description of the tasks and evaluations.

### C.1 LONG-HORIZON TASKS

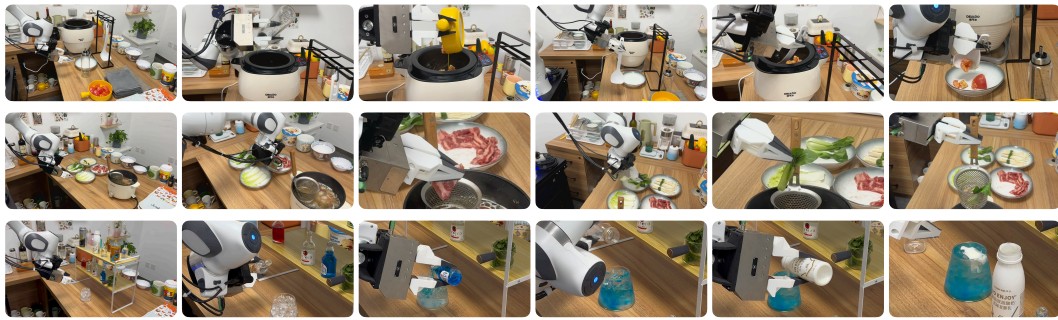

Figure 8: **Execution processes of three long-horizon tasks**: `Tomato-Egg`, `Hotpot`, and `Cocktail` (exemplified by `Mountain Fuji` preparation).

Fig. 8 shows the complete execution progress of the three long-horizon tasks. Detailed descriptions of these tasks are as follows:

1) `Tomato-Egg`: The robot first pours oil, then tomato and egg liquid into a cooking machine. Once cooking is finished, the robot picks up a spoon hanging on a rack, scoops out the tomato-egg scramble, transfers it onto a plate, and finally places the spoon into the cooking machine. We observe that sometimes the robot fails to grip the oil bottle firmly enough, causing it to slip from the gripper. We collect dedicated recovery data for re-grasping the oil bottle more securely after it has slipped. This enables the robot to automatically perform this recovery if it encounters a bottle slip during testing. We collect 200 robot demonstrations for this task.

2) `Hotpot`: Four plates containing beef, green bok choy, enoki mushrooms, and cabbage are placed on a table with randomized relative positions. A hotpot with a strainer is positioned to the right of the plates. For each test, the human instructs the robot to dip beef and one type of vegetable. The robot must accurately pick up the ingredients sequentially, place them in the strainer, wait for them to cook, and then lift the strainer. Notably, for OneTwoVLA and the dual-system approach, in 10 of the experiments, the initial instruction is only to dip the beef. While waiting for the beef to cook, the human interacts with the robot saying,"Could you also dip another vegetable for me?", requiring the robot to ask, "Sure! Would you like green bok choy, enoki mushrooms, or cabbage?" Following the human's specification, the robot then proceeds to dip the requested vegetable. This interaction step is omitted for $\pi_0$ due to its lack of text output capabilities. Furthermore, we observe instances where the robot fails to grasp the strainer. To address this, we specifically collect recovery data for correcting misaligned grasps. This enables the robot to automatically perform this recovery if it fails to pick up the strainer during testing. We collect 600 robot demonstrations for this task.

3) `Cocktail`: The robot is instructed to prepare one of three cocktails: `Mojito`, `Mountain Fuji`, or `Vodka Sunrise`. Each cocktail requires pouring 3-4 different ingredients. For OneTwoVLA and the dual-system approach, in 10 trials, the initial human instruction is general: "Make me a cocktail." The robot must clarify by asking: "Which cocktail would you like?", and then proceed based on the human's specific cocktail choice. This interaction step is again omitted for $\pi_0$. Additionally, during 3 separate `Vodka Sunrise` trials, the human interrupts with, "I don't want orange vodka, I want lemon-flavored one," requiring the robot to put down the orange vodka and pick up lemon vodka instead. We collect 100 robot demonstrations for each type of cocktail, totaling 300 demonstrations.

## C.2 GENERALIZABLE PLANNING TASKS

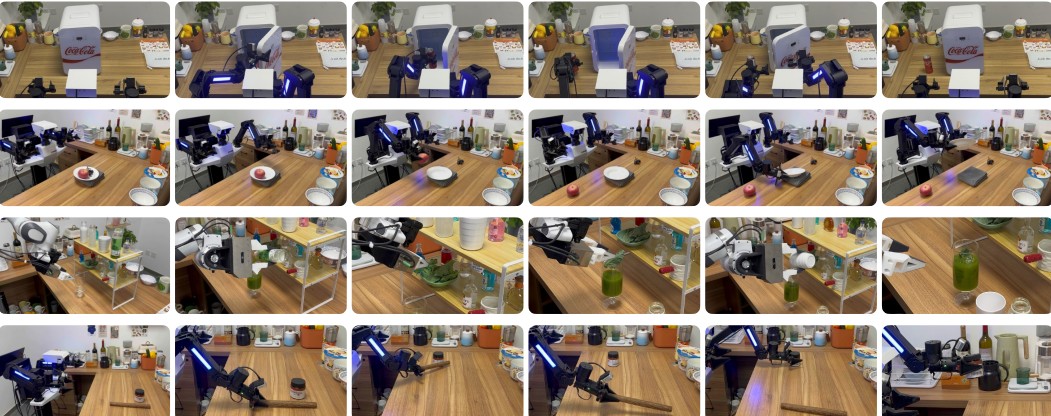

Figure 9: **Execution processes of four generalizable planning tasks**: `Get Icy Cola`, `Empty Plate`, `Prepare Drinks` (exemplified by `kale juice` preparation) and `Tool Use`.

We collect 2,000 robot demonstrations using the single-arm Franka system and dual-arm ARX system. Each demonstration belongs to one category of atomic skill, including pick, place, move, open, close, and pour. The task instructions and corresponding reasoning contents for these demonstrations focus on short-horizon atomic skills. Training solely on this data limits the model's generalizable long-horizon planning capabilities. OneTwoVLA overcomes this limitation through co-training with our synthesized embodied reasoning-centric vision-language data, which equips it to generalize to previously unseen tasks. Fig. 9 shows the complete execution progress of these unseen tasks. Detailed descriptions of these tasks are as follows:

1) `Get Icy Cola`: The instruction is "Get me a can of icy cola." The challenge is that a cola can is not directly visible in the scene. The robot must infer that "icy cola" implies the cola is stored in the fridge and therefore plan the necessary steps to open the fridge, locate the cola, and retrieve it.

2) `Empty Plate`: The instruction is "Pass me an empty plate". However, the plate in the scene is not empty, as it contains apples and grapes. The robot needs to remove each fruit from the plate before finally picking up the empty plate.

3) `Tool Use`: The instruction is "Pick up the cocoa powder can, which is out of reach". The primary difficulty here is that the target object is not within the robot's direct reach. The robot must recognize the need for a tool (a nearby stick), plan to first grasp the stick, use it to sweep the distant cocoa powder can within reach, and only then proceed to pick up the can.

4) `Prepare Drinks`: The robot needs to plan and prepare appropriate drinks based on user intent: such as `coconut latte` for "Help me stay awake," `kale juice` for "I want something healthy," and a `blue mood` cocktail for "I'm feeling down." This task requires scene-aware user intent understanding capability.

## C.3 VISUAL GROUNDING TASKS

Task descriptions can be found in Sec. 4.4. In the `Single-Env` setting, each robot demonstration is paired with 11 instruction-reasoning pairs. These instructions refer to target objects using their names (2 instances), spatial relationships (3 instances), attributes (3 instances), or semantic features (3 instances). In the `Open-World` setting, each demonstration includes a total of 17 instruction-reasoning pairs, broken down as 2 using direct names, 5 using spatial relationships, 5 using attributes, and 5 using semantic features. All instruction–reasoning pairs are first generated with Gemini 2.5 Pro and then verified by human annotators.

During testing, we evaluate each method 40 times in both settings. This consists of 10 tests for each reference type. Table 4 presents the experimental results broken down by these four types.

Here we list the objects used in visual grounding tasks. The `Single-Env` task uses four objects: blue cube, eggplant toy, coconut water bottle, and black mouse. For the `Open-World` task evaluation, we use the following objects (shown in Fig. 10):

1) 5 objects seen in robot data: flower, mouse, cardholder, tissue, and glasses case.

2) 10 objects unseen in robot data but present in synthetic vision-language data: globe, teddy bear, straw hat, binoculars, trowel, croissant, map, magnifying glass, VR headset, lantern.

3) 5 objects unseen in either dataset: GoPro, Sprite, Starbucks Coffee, HDMI cable, Captain America model.

Fig. 11 displays the 16 training environments for the `Open-World` task, while Fig. 12 shows the 8 evaluation environments.

| | Single-Env | | | | | Open-World | | | | |
|---|---|---|---|---|---|---|---|---|---|---|
| | Name | Spatial | Attribute | Semantic | Total | Name | Spatial | Attribute | Semantic | Total |
| OneTwoVLA-VL | 10/10 | 8/10 | 8/10 | 9/10 | 35/40 | 8/10 | 6/10 | 7/10 | 8/10 | 29/40 |
| OneTwoVLA | 10/10 | 5/10 | 8/10 | 8/10 | 31/40 | 2/40 | 0/10 | 1/10 | 0/10 | 3/40 |
| $\pi_0$ | 2/10 | 0/10 | 0/10 | 0/10 | 2/40 | 1/10 | 0/10 | 0/10 | 0/10 | 1/40 |

Table 4: **Experimental results for the visual grounding tasks.** Results are broken down by the four instruction reference types: direct names, spatial relationships, object attributes, and semantic features.

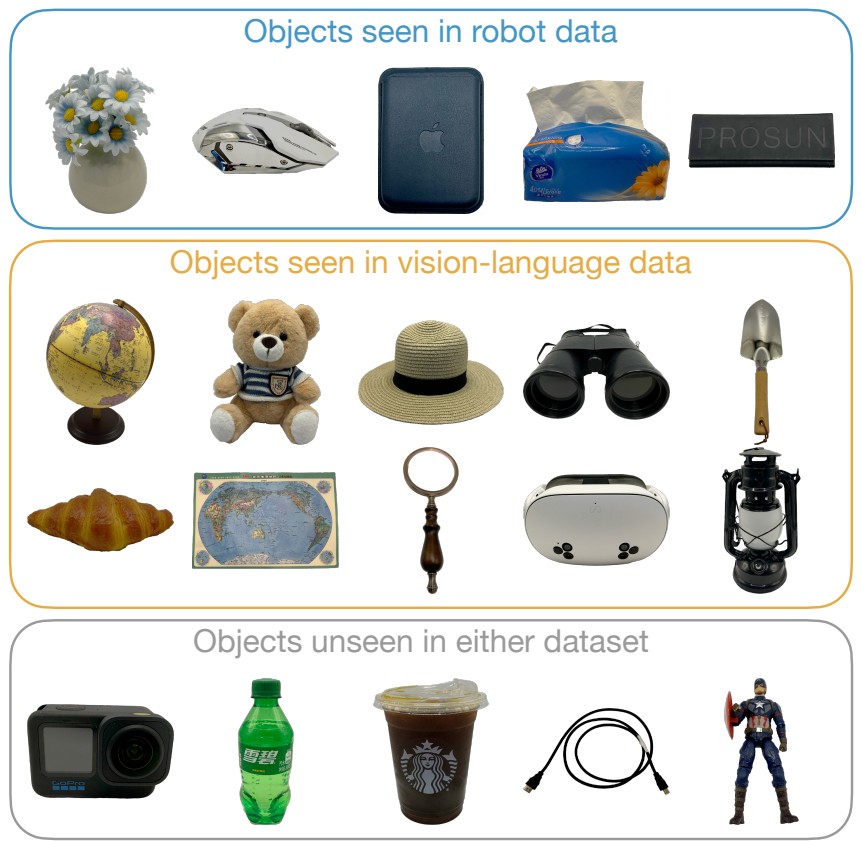

Figure 10: **Objects for `Open-World` task evaluation.**

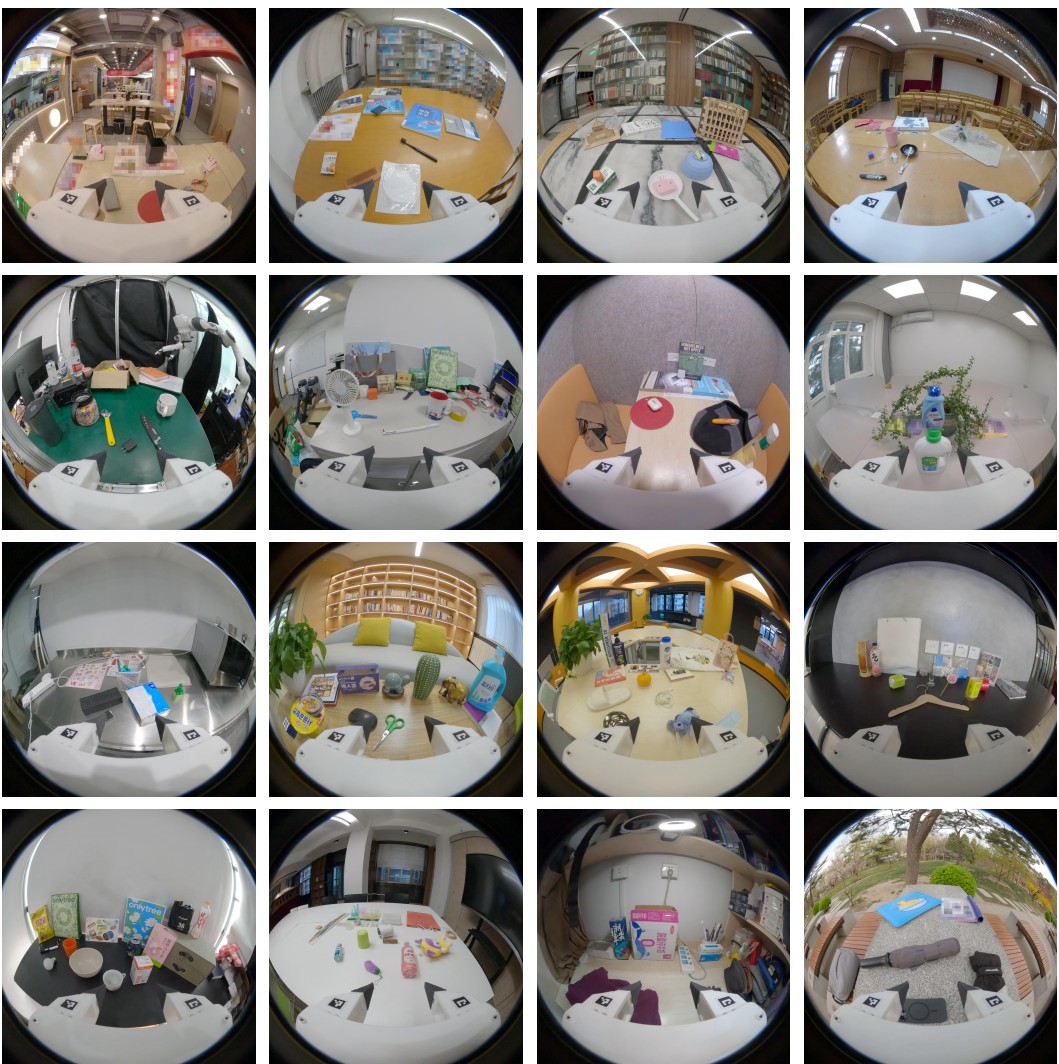

Figure 11: **Training environments for `Open-World` visual grounding task.**

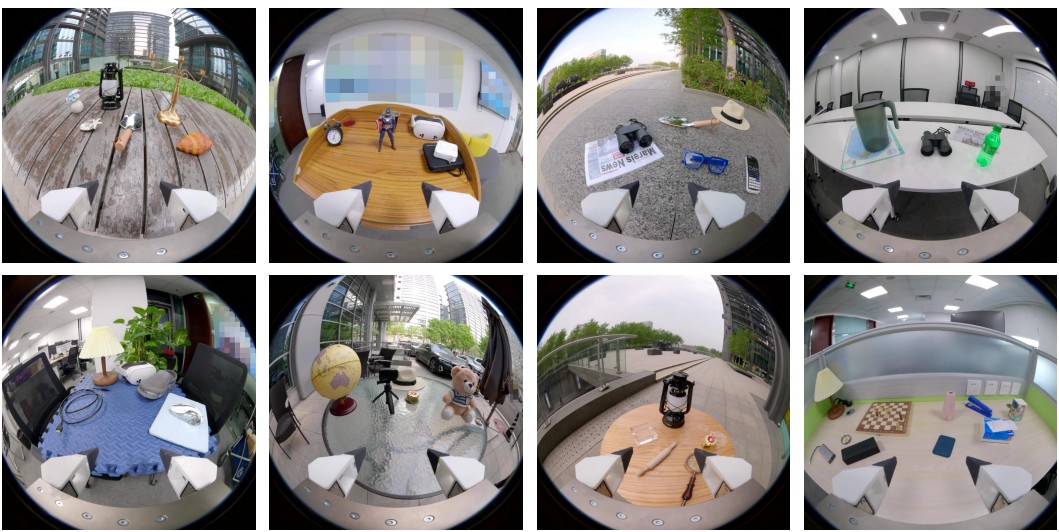

Figure 12: **Evaluation environments for `Open-World` visual grounding task.**

## D    MORE REASONING EXAMPLES

Detailed OneTwoVLA reasoning examples during task execution are presented in this section. These include examples for long-horizon task planning (Table 5), generalizable planning (Table 6), error detection and recovery (Table 7), natural human-robot interaction (Table 8), `Single-Env` visual grounding (Table 9), and `Open-World` visual grounding (Table 10).

We provide prompt templates for annotating reasoning intervals (Fig. 13) and for generating reasoning content (Fig. 14).

Our template can readily extend to error detection/recovery and human–robot interaction (HRI) via a plan-augmentation step. Given the original high-level plan $P = (p_1, \ldots, p_K)$, we construct an augmented plan $\tilde{P}$ by inserting explicit substeps that represent anticipated recovery events and interaction turns (e.g., "detect failure," "retract," "retry," "ask question," "receive answer"). We use $\tilde{P}$ to prompt Gemini in both stages—(i) interval annotation and (ii) reasoning-content generation—so that reasoning intervals and their fields align with these events.

**Error detection and recovery.** Using $\tilde{P}$ encourages the annotator to place reasoning intervals around failure and recovery micro-steps (e.g., a failed grasp followed by a retraction). During assembly, however, we map back to the original plan $P$. The reasoning tuple remains $(D_j, P, H_j, X_j)$, where $H_j = \mathrm{render}(p_1, \ldots, p_j)$ over $P$. If a failure occurs on $p_k$, we set the next step to a retry action conditioned on $p_k$ (e.g., $X_j = \mathrm{retry}(p_k)$), optionally preceded by a retraction, without altering $P$ itself.

**Human–robot interaction.** In $\tilde{P}$ we insert explicit interaction substeps, such as the robot asking a question and the human providing an answer (e.g., "Which cocktail should I make?" → "Mojito"). During assembly, we extend the reasoning tuple to $(D_j, P_j, H_j, X_j, Q_j, A_j)$, where $Q_j$ is the robot's question and $A_j$ is the human's answer (empty if no interaction). The plan is updated around the dialogue: before the interaction we use a provisional plan $P_j^{\mathrm{pre}}$, and after observing $A_j$ we synthesize a conditioned plan $P_j^{\mathrm{post}}$ that replaces the remaining substeps; thus $P_j = P_j^{\mathrm{pre}}$ before the dialogue and $P_j = P_j^{\mathrm{post}}$ after. The historical summary $H_j$ records the newly acquired information (i.e., $A_j$), and the next step $X_j$ is conditioned on $A_j$ (e.g., proceed to gather ingredients for a Mojito). Interaction boundaries are marked as reasoning intervals during annotation.

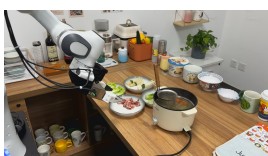

**Instruction**: Dip the beef and the green bok choy.
**Reasoning**:
Scene description: The beef is located on the bottom-right plate. The green bok choy is on the top-right plate. The pot is in the center-right area of the table and contains a hotpot strainer.
Plan: 1. Put the beef into the hotpot strainer. 2. Put the green bok choy into the hotpot strainer. 3. Wait for the ingredients to cook and then pick up the hotpot strainer.
What I have done: Nothing.
Now I need to: Put the beef into the hotpot strainer.

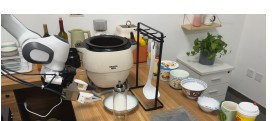

**Instruction**: Make me a plate of tomato-egg scramble.
**Reasoning**:
Scene description: The oil bottle is located at the front of the table. The cooking machine is on the left side of the table.
Plan: 1. Pour the oil into the cooking machine. 2. Pour the tomatoes and egg liquid into the cooking machine. 3. Wait for the cooking process to finish, then pick up the spoon. 4. Scoop out the tomato-egg scramble, transfer it onto the plate, and place the spoon into the cooking machine.
What I have done: Nothing.
Now I need to: Pour the oil into the cooking machine.

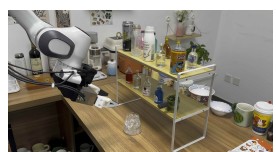

**Instruction**: Make me a cup of `Mountain Fuji` cocktail.
**Reasoning**:
Scene description: The orange-flavored vodka is the second from the left on the bottom shelf. The Blue Curacao is the second from the right on the bottom shelf. The lemon juice is the second from the left on the top shelf. The yogurt is located in the middle of the top shelf.
Plan: 1. Pour the orange-flavored vodka into the glass. 2. Pour the Blue Curacao into the glass. 3. Pour the lemon juice into the glass. 4. Pour the yogurt into the glass.
What I have done: Nothing.
Now I need to: Pour the orange-flavored vodka into the glass.

Table 5: **Reasoning examples for long-horizon task planning.**

| | |
|---|---|
| 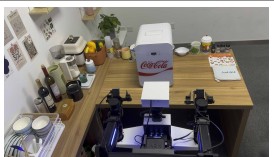 | **Instruction**: Give me an icy cola.
**Reasoning**:
Scene description: The cola is out of sight, maybe in the fridge.
Plan: 1. Open the fridge door. 2. Pick up the icy cola bottle and place it on the table. 3. Close the fridge door. |
| 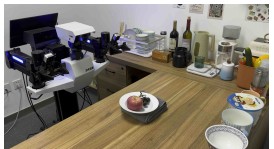 | **Instruction**: Pass me an empty plate.
**Reasoning**:
Scene description: The grapes and apple are on the plate.
Plan: 1. Pick up the grapes and place them on the table. 2. Pick up the apple and place it on the table. 3. Pick up the empty plate. |
| 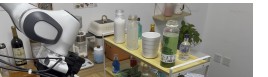 | **Instruction**: I'm exhausted — can you make me a drink to keep me awake?
**Reasoning**:
I need to make a `coconut latte`.
Plan: 1. Add thick coconut milk to the coffee cup. 2. Pour espresso into the cup. 3. Drop in ice cubes. |
| | **Instruction**: I've been eating clean and need a nutritious drink.
**Reasoning**:
I need to make a glass of `kale juice` with some kale leaves and ice cubes.
Plan: 1. Pour kale juice into the cup. 2. Put a kale leaf in the cup. 3. Add ice cubes to the cup. |
| | **Instruction**: I'm feeling kind of down today. Can you fix me a drink?
**Reasoning**:
I need to make a `Blue Mood` cocktail.
Plan: 1. Pour in white rum. 2. Add Blue Curacao. 3. Mix in lemon juice. 4. Drop in ice cubes. |
| 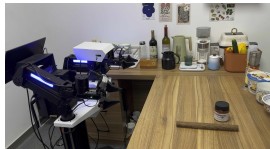 | **Instruction**: Pick up the cocoa powder bottle that is out of reach.
**Reasoning**:
Scene description: The cocoa power bottle is out of reach. There is a wooden stick nearby.
Plan: 1. Grasp the stick. 2. Sweep the cocoa powder bottle closer with the stick and put down the stick. 3. Pick up the cocoa powder bottle. |

Table 6: **Reasoning examples for generalizable planning.**

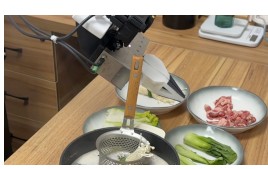

**Instruction**: Dip the beef and the enoki mushrooms.
**Reasoning**:
Scene description: Some of the beef and the green bok choy are in the hotpot strainer. *But I fail to grasp the hotpot strainer due to misalignment.*
Plan: 1. Put the beef into the hotpot strainer. 2. Put the enoki mushrooms into the hotpot strainer. 3. Wait for the ingredients to cook and then pick up the hotpot strainer.
What I have done: 1. Put the beef into the hotpot strainer. 2. Put the enoki mushrooms into the hotpot strainer.
Now I need to: *Retract, reposition to align with the strainer, and try grasping again.*

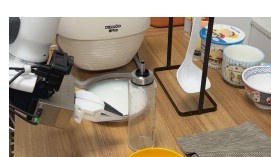

**Instruction**: Make me a plate of tomato-egg scramble.
**Reasoning**:
Scene description: The oil bottle is located at the front of the table. The cooking machine is on the left side of the table. *But I discover that the oil bottle slips from the gripper.*
Plan: 1. Pour the oil into the cooking machine. 2. Pour the tomatoes and egg liquid into the cooking machine. 3. Wait for the cooking process to finish, then pick up the spoon. 4. Scoop out the tomato-egg scramble, transfer it onto the plate, and place the spoon into the cooking machine.
What I have done: Nothing.
Now I need to: *Retry to grasp the oil bottle with increased firmness.*

Table 7: **Reasoning examples for error detection and recovery.**

| | |
|---|---|
| | **Instruction**: Dip the beef for me.
**Human**: *Could you also dip another vegetable for me?*
**Robot**: *Sure! Would you like green bok choy, enoki mushrooms, or cabbage?*
**Human**: *I want some green bok choy.*
**Reasoning**:
Scene description: Some of the beef is in the hotpot strainer. *The green bok choy is on the top-left plate.*
Plan: 1. Put the beef into the hotpot strainer. *2. Put the green bok choy into the hotpot strainer.* 3. Wait for the ingredients to cook and then pick up the hotpot strainer.
What I have done: 1. Put the beef into the hotpot strainer.
Now I need to: *Put the green bok choy into the hotpot strainer.* |
| | **Instruction**: Make me a cup of `Vodka Sunrise`.
**Human**: *I don't want orange vodka, I want lemon-flavored one.*
**Reasoning**:
Scene description: The grenadine syrup and orange juice have been poured into the glass. *The lemon-flavored vodka is the second from the right on the top shelf.*
Plan: 1. Pour the grenadine syrup into the glass. 2. Pour the orange juice into the glass. *3. Put down the orange vodka, and retrieve the lemon vodka.*
What I have done: 1. Pour the grenadine syrup into the glass. 2. Pour the orange juice into the glass.
Now I need to: *Put down the orange vodka, and retrieve the lemon vodka.* |

Table 8: **Reasoning examples for natural human robot interaction.**

| | |
|---|---|
| 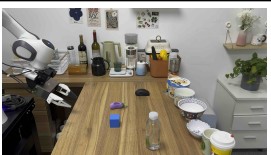 | **Instruction**: Pick up the black mouse.
**Reasoning**:
I need to pick up the black mouse, which is to the left back of the eggplant, has a smooth, arching shape that fits the palm, and is a key component of a typical office workstation. |
| 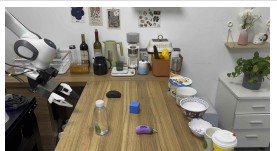 | **Instruction**: Pick up the object which is to the left back of the coconut water bottle.
**Reasoning**:
I need to pick up the blue cube, which is identified by its blue shade, and is a solid form useful for creating stable structures. |
| 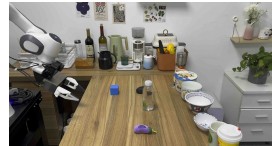 | **Instruction**: Pick up the object which exhibits a purple coloration.
**Reasoning**:
I need to pick up the eggplant, which is to the right of the coconut water bottle, and is a common ingredient in ratatouille. |
| 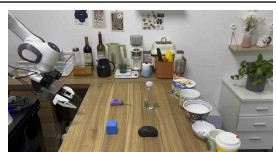 | **Instruction**: Pick up the object which is a packaged drink.
**Reasoning**:
I need to pick up the coconut water bottle, which is to the left back of the blue cube, and has a cylindrical form. |

Table 9: **Reasoning examples for `Single-Env` visual grounding.**

| | |
|---|---|
| 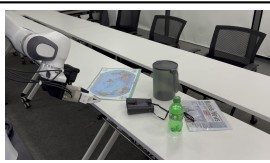 | **Instruction**: Bring me the Sprite.
**Reasoning**:
I need to pick up green Sprite, which is a model of a drink, located on the right side of the table. |
| 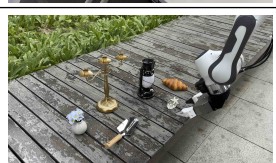 | **Instruction**: Give me the object behind the seashell.
**Reasoning**:
I need to pick up the lantern to the left of the croissant, which is black with a metal structure, and holds light and can be carried. |
| 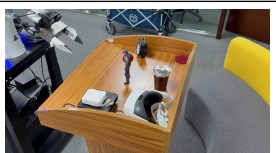 | **Instruction**: Hand me the object which is black and round.
**Reasoning**:
I need to pick up the alarm clock sitting on the left side of the table. |
| 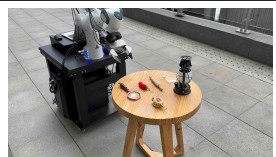 | **Instruction**: I need something to get a closer look at this fine print.
**Reasoning**:
I need to pick up the magnifying glass, which is to the right of the red pepper, and has a wooden handle and a circular lens. |

Table 10: **Reasoning examples for `Open-World` visual grounding.**

You are given:
A step-by-step plan for a tabletop robot task, recorded by a wrist-mounted camera.
A set of sampled frames from the video, named by integer frame IDs.
Each image is introduced by a text line "Frame <id>" followed by the image itself.

Goal:
Annotate "boundary intervals" between consecutive steps in the plan. A boundary interval is a contiguous range of frames [start_id, end_id] such that:
1) From the visual evidence, the previous step has clearly been completed.
2) The next step has not yet started.
3) The tabletop is substantially visible across the interval (i.e., a relatively complete view of the workspace surface; avoid frames where the table is mostly occluded by the robot or out of view).
4) Gripper-state criterion for all selected intervals: the gripper is open, the gripper holds no object (empty), and it is positioned relatively far from the tabletop (i.e., retracted or clearly above the surface rather than near contact). Prefer intervals where this is directly visible.

Additionally:
Typical step structure involves: pick a tool/object, perform an operation, then place/put the tool elsewhere. Favor boundary intervals after "place/release" of the prior step and before the approach to the next target, where the gripper is open, empty, and away from the tabletop.
If the gripper is temporarily occluded or out of view, accept frames where other cues strongly indicate open/empty/away (e.g., no contact, no object in jaws across adjacent frames). If not confirmable, keep the interval minimal and lower the confidence, but still follow the schema.
Include a pre-task boundary interval that begins at frame 0 (implicit) and ends just before Step 1 visibly begins, while satisfying tabletop visibility and the gripper-state criterion.
Include a post-task boundary interval that starts right after the final step is complete and ends at the last frame (implicit), while satisfying tabletop visibility and the gripper-state criterion.
Intervals must be non-overlapping, ordered, and as long as possible while respecting the rules above.
Prefer intervals with at least 2–3 frames where possible. If only a single frame satisfies the conditions, use a single-frame interval [k, k].
If some step transition has no frames that both (a) show previous-step-complete and next-step-not-started and (b) maintain tabletop visibility and (c) meet the gripper-state criterion, pick the closest visually justifiable frame(s) and keep the interval minimal, but never violate chronological order.

Assumptions and cues:
"Previous step completed" cues can include: the target object placed in its goal pose, gripper released/open, robot no longer manipulating that object, or the scene stabilized.
"Next step not started" cues can include: the gripper has not yet approached/touched the next target, no new object interaction has begun, or no evident motion initiating the next step.
The camera is on the wrist; favor frames where the tabletop surface is broadly visible and not heavily occluded by the manipulator.

Input you receive:
Step plan (numbered 1..N).
A list of frames; each is introduced by the line "Frame <id>" followed by its image. Frame IDs are chronological and unique. The first frame ID is 0; the last is given.

Output format (return ONLY this JSON; no extra text):
{"version":"v1","total_frames":"","steps":["<step 1 text>","<step 2 text>","..."],"intervals":
[{"label":"pre_step_1","end":"","tabletop_visible":true,"confidence":"<float 0..1>"},
{"label":"between_step_1_2","start":"","end":"","tabletop_visible":true,"confidence":"<float 0..1>"},
{"label":"...","start":"","end":"","tabletop_visible":true,"confidence":"<float 0..1>"},
{"label":"between_step_(N-1)_N","start":"","end":"","tabletop_visible":true,"confidence":"<float 0..1>"},
{"label":"post_step_N","start":"","tabletop_visible":true,"confidence":"<float 0..1>"}]}

Figure 13: **Prompt template for annotating reasoning interval.**

You are given:
A central frame image (the current interval's keyframe).
The task's high-level plan as an ordered list of substeps P = (p1, ..., pK).
The list of completed substeps up to the current interval.
The next substep (or "Task Finished" if the task is finished).

Your goal:
1) Produce the "scene description" for the current interval focusing strictly on task-relevant objects for the given plan and substeps.
2) Output must be valid JSON only, matching the schema below.

Position frame of reference:
"front" = nearest edge of the table to the camera/viewer.
"back" = farthest edge of the table from the camera/viewer.
"left/right" = viewer's left/right.
You may also use "center/middle," "front-left," "front-right," "back-left," "back-right," and proximity phrases like "near the left edge," "near center," "near the back edge."

Rules:
Write a concise scene summary (2–4 sentences).
List only task-relevant objects on the table (tools, containers, parts, targets, intermediates).
One sentence per object describing its absolute position on the table using the frame above.
Do not hallucinate; if uncertain or partially occluded, lower confidence and note the uncertainty.
If multiple similar objects exist, disambiguate with indices (e.g., "blue block #1", "blue block #2") based on spatial layout.
If an object is not on the table or not visible, omit it from the objects list.
Use brief, literal attributes (e.g., color, material, state like "open/closed," "full/empty").
Output must be valid JSON only. Do not include any extra text or formatting outside JSON.

Inputs:
TASK_NAME: {TASK_NAME}
HIGH_LEVEL_PLAN (ordered): {HIGH_LEVEL_PLAN_AS_LIST}
COMPLETED_SUBSTEPS (up to current interval): {COMPLETED_SUBSTEPS_AS_LIST}
NEXT_SUBSTEP: {NEXT_SUBSTEP_OR_DONE}
IMAGE: [central frame image provided as input]

Output JSON schema:
{ "scene_summary": "string, 2–4 sentences describing the visible scene and its relevance to the task.", "objects": [ { "name": "string, concise object name (e.g., 'red mug', 'blue block #1')", "attributes": ["string", "..."], "position_sentence": "string, exactly one sentence stating the object's absolute position on the table (front/back/left/right/center + optional proximity/edge terms).", "position_tags": ["front| back|left|right|center", "optional tags like 'front-left', 'near-left-edge', 'near-back-edge'"], "confidence": 0.0 } ], "uncertain_observations": ["string notes about any ambiguities, occlusions, or visibility issues (optional; omit if none)"] }

Figure 14: **Prompt template for generating reasoning content (scene descriptions).**

# E  SYNTHETIC VISION-LANGUAGE DATA EXAMPLES

Our 16,000 synthetic images are entirely annotated by Gemini 2.5 Pro, without any human intervention. For 6,000 of these images, we generate visual grounding tasks. Each of these images is annotated with 17 instruction-reasoning pairs, with the instructions referring to objects using their direct names (2 instances), spatial relationships (5 instances), attributes (5 instances), and semantic features (5 instances). For the remaining 10,000 images, we annotate a long-horizon planning task along with a corresponding high-level, step-by-step plan for task completion. We also attempt to use GPT-4o for annotating our synthetic images but find its spatial understanding to be weak. We therefore use Gemini 2.5 Pro, which demonstrates strong spatial reasoning capabilities.

We present illustrative examples synthesized by our embodied reasoning-centric visual-language data synthesis pipeline. Table 12 and Table 13 show samples of synthesized data for visual grounding and long-horizon tasks, respectively, each including textual descriptions of tabletop layouts, synthesized images corresponding to these descriptions, and the accompanying instruction-reasoning pairs (for visual grounding example, we only show one pair for each of the four reference types). Fig. 15 illustrates the effects of applying fisheye distortion or compositing a robot gripper with adaptive brightness to the synthetic images.

Moreover, Fig. 16, 17, and 18 detail the specific prompts used with Gemini 2.5 Pro throughout our pipeline: Fig. 16 shows the prompt for generating diverse tabletop descriptions, while Fig. 17 and 18 illustrate the prompts for generating visual grounding and long-horizon task instructions and their associated reasoning, respectively.

**Data Quality.** We randomly sample 50 instances from the synthetic dataset and identify the following types of errors: 1) *Wrong Image*: the presence of hallucinated objects or implausible physics; 2) *Wrong Text*: reasoning that contradicts the corresponding image or common sense. If the generated image is incorrect, we do not evaluate its text. The results are summarized in Table 11.

| Wrong Image | Wrong Text | All Correct |
|:---:|:---:|:---:|
| 2 | 8 | 40 |

Table 11: **Frequency of errors and fully correct cases in 50 randomly sampled synthetic data instances.**

**Tabletop layout description**: The entire scene is sharp and focused. From a 45-degree overhead view, observe a medium-sized, rectangular driftwood table with a natural, uneven texture. On it are displayed seashells, a coral sculpture, a nautical map under a magnifying glass, a vintage lantern, and a model sailboat, backed by a beach house and ocean view.

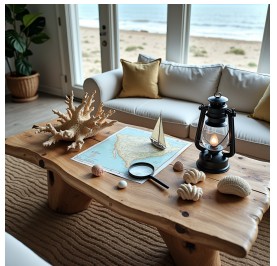

**Direct instruction**: Can you pass me the map?
**Reasoning**: I need to pick up the map, located towards the center of the table.

**Spatial instruction**: Get me the object sitting on the back right area of the paper item.
**Reasoning**: I need to pick up the miniature sailboat model, which is placed on the map near the top right corner of the map.

**Attribute instruction**: I need the item with the complex branching structure.
**Reasoning**: I need to pick up the large piece of coral positioned at the left side of the table.

**Semantic instruction**: Could you pass me the device used for seeing small details much larger?
**Reasoning**: I need to pick up the magnifying glass situated on the map near the bottom right corner of the map.

**Tabletop layout description**: The entire scene is sharp and focused. From a 45-degree overhead view, observe a large, oval, stone table with a rough texture, set outdoors in a mountain retreat. On it, a hearty meal of grilled trout, wild rice, steamed vegetables, a jug of water, and a lantern. The rugged outdoor setting is enhanced by the surrounding pine trees and the fading twilight.

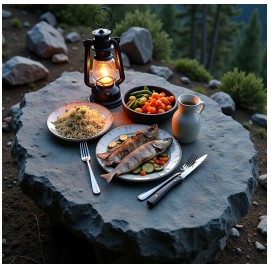

**Direct instruction**: Can you hand me the lantern?
**Reasoning**: I need to pick up the black lantern situated at the back left of the rock table.

**Spatial instruction**: Pass me the item directly to the left of the grilled trouts.
**Reasoning**: I need to pick up the fork sitting to the immediate left of the grilled trouts.

**Attribute instruction**: Please give me the plate with the long, silvery food item.
**Reasoning**: I need to pick up the plate containing the two cooked fish positioned in front of the vegetable bowl.

**Semantic instruction**: Hand me that grain-based side dish, please.
**Reasoning**: I need to pick up the plate of wild rice located at the back left, to the left of the vegetable bowl.

**Tabletop layout description**: The entire scene is sharp and focused. From a 45-degree overhead view, observe a rectangular maple table with a honey finish, placed in a sunny conservatory. A watering can, a terracotta pot with blooming flowers, a gardening book, gloves, and a small birdhouse are charmingly scattered on it. Glass panels reveal a lush garden outside.

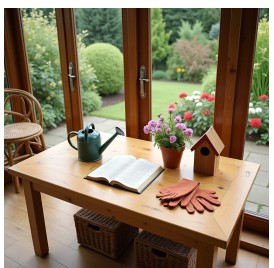

**Direct instruction**: Can you pass me the watering can?
**Reasoning**: I need to pick up the green watering can positioned on the back left side of the table.

**Spatial instruction**: I want the item located on the front right side of the table.
**Reasoning**: I need to pick up the pair of gardening gloves, situated on the front right area of the table surface.

**Attribute instruction**: I need the object that's open and has pages with text.
**Reasoning**: I need to pick up the book, which is open and located between the watering can and the brown gloves.

**Semantic instruction**: Please pass me the item that could provide shelter for small birds.
**Reasoning**: I need to pick up the wooden birdhouse, sitting on the back right side of the table.

Table 12: **Examples of synthetic vision-language data for visual grounding tasks.**

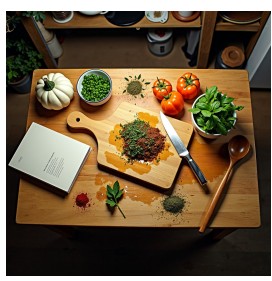

**Tabletop layout description**: The entire scene is sharp and focused. From a 45-degree overhead view, directly facing the cluttered table, observe a rustic wooden surface covered in vibrant fresh ingredients like tomatoes, peas, herbs, and spices. A cutting board, knife, spoon, and book are laid out, bathed in bright light, against a backdrop of softly visible shelves with plants.

**Instruction**: Prepare a Fresh Herb and Tomato Relish.
**Reasoning**:
Plan: 1. Chop the tomatoes using the knife and cutting board. 2. Chop the fresh basil and parsley. 3. Combine the chopped tomatoes, basil, and parsley in a bowl. 4. Add the green peas and the dried herbs and spices from the piles on the board to the bowl. 5. Mix all ingredients thoroughly using the wooden spoon.

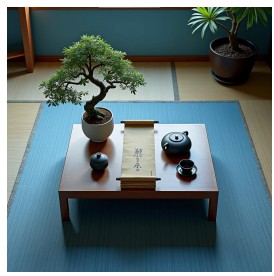

**Tabletop layout description**: The entire scene is sharp and focused. From a 45-degree overhead view, directly facing the cluttered table, observe a low Japanese tea table with a woven tatami mat beneath it, showcasing a tea set, a small bonsai tree, and a scroll. The scene plays out indoors, in a serene, minimalist room, emphasizing peaceful tranquility.

**Instruction**: Prepare and serve tea.
**Reasoning**:
Plan: 1. Pick up the teacup and place it next to the teapot. 2. Fill the teapot with hot water. 3. Pour tea from the teapot into the teacup. 4. Place the teapot on the table. 5. Serve the teacup to the guest.

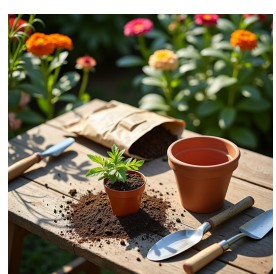

**Tabletop layout description**: The entire scene is sharp and focused. From a 45-degree overhead view, directly facing the cluttered table, observe a small potted plant beside an empty terracotta pot. A soil bag, a silver trowel, and various tools are scattered across the worn wooden surface. The scene takes place outdoors, in a garden full of flowers, bathed in warm afternoon light.

**Instruction**: Repot the small plant into the larger terracotta pot.
**Reasoning**:
Plan: 1. Remove the plant from its current small pot. 2. Add soil from the bag into the bottom of the larger terracotta pot using the trowel. 3. Place the plant in the center of the larger pot. 4. Fill the remaining space in the larger pot with soil from the bag using the trowel. 5. Use the trowel to gently firm the soil around the base of the plant.

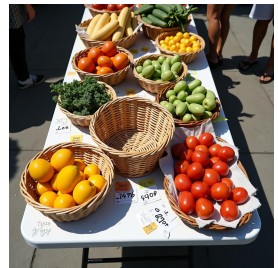

**Tabletop layout description**: The entire scene is sharp and focused. From a 45-degree overhead view, directly facing the cluttered table, observe a white folding table at an outdoor market. Assorted fruits, vegetables, and price tags cover the table. The bright, natural light enhances the colors of the produce.

**Instruction**: Prioritize fresh produce to boost your daily vitamin intake.
**Reasoning**:
Plan: 1. Pick up some tomatoes and place them in the basket. 2. Pick up some lemons and place them in the basket. 3. Pick up some oranges and place them in the basket. 4. Carry the basket to checkout.

Table 13: **Examples of synthetic vision-language data for long-horizon tasks.**

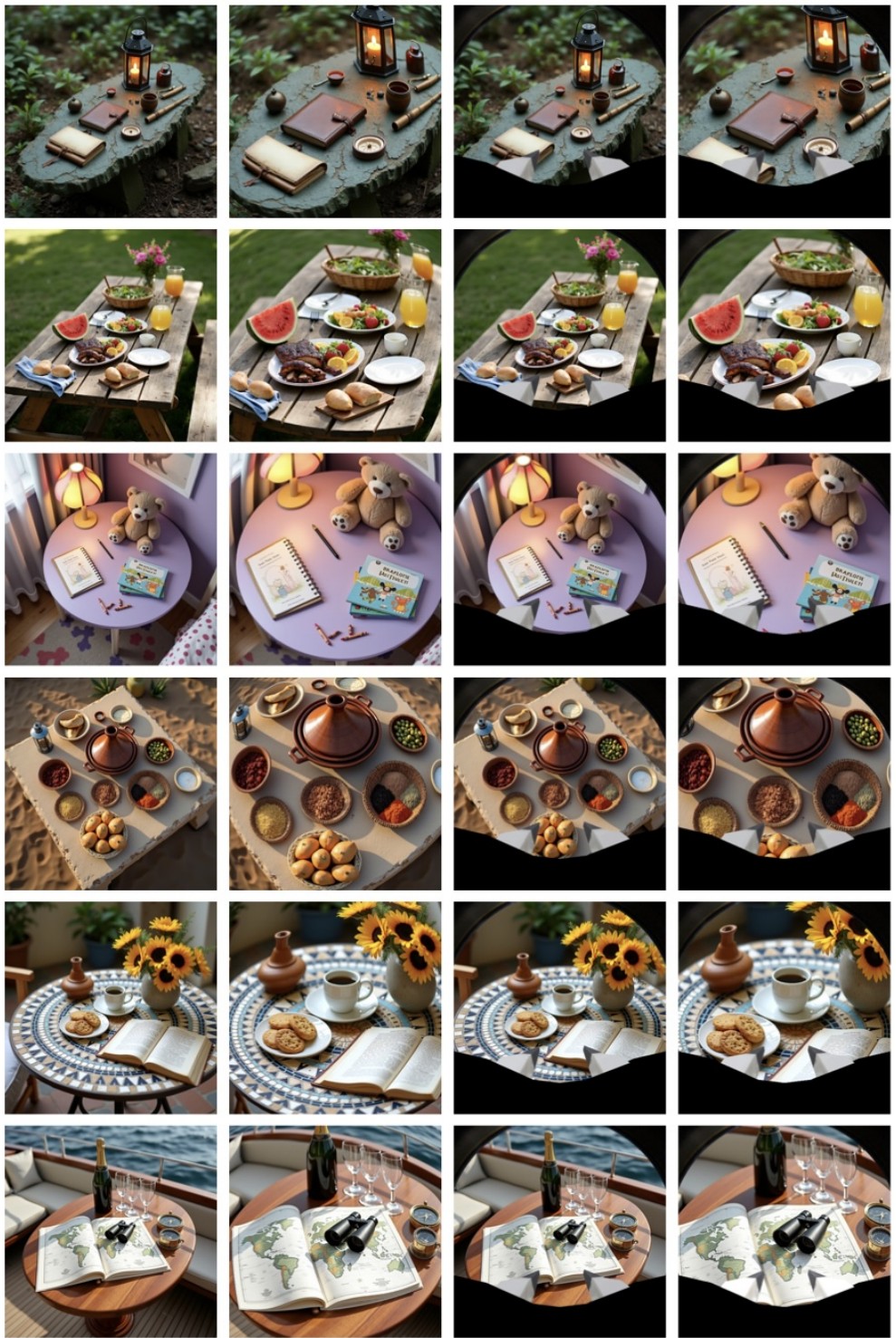

Figure 15: **Augmentations for our synthetic images.** From left to right: original synthetic images, synthetic images with fisheye distortion, synthetic images with a robot gripper composited with adaptive brightness, and synthetic images with both fisheye distortion and compositing a robot gripper with adaptive brightness.

Create 30 detailed 50-word prompts that describe scenes from a 45-degree top-down view of a table. The table should have a clear description of its shape, size, texture, and color. On the table, place around five objects, describing each object in detail and their positions relative to each other (e.g., object A is placed above object B). The background and environment should be clearly defined, either indoor or outdoor, and the scene should be rich in detail. Ensure there is no blurriness or out-of-focus areas, and the lighting and atmosphere should enhance the realism.

Please ensure each of the following prompts is unique and creatively different, varying the table, objects, environment (like indoor or outdoor), lighting, and overall atmosphere.

Each prompt should start with "The entire scene is sharp and focused. From a 45-degree overhead view, observe ...", followed by a description of the table's **COLOR** (this could be diverse across different prompts), shape, texture, size, etc.

Use the following format to separate each prompt:
**START Prompt <Prompt ID>**
[Detailed description of the scene]
**END Prompt <Prompt ID>**

Repeat this process for 30 distinct prompts. Request to generate all at once.

Figure 16: **Prompt used to generate tabletop descriptions.**

In the provided image, you will notice several items placed on a table. Your task is to come up with 17 different instructions based on these items. These tasks will be categorized into three types based on object properties: spatial, semantic, and attribute.

Spatial pertains to the object's position in space (e.g., on top of the plate, to the right of the book, or at the bottom right corner of the table).

Semantic refers to the object's general, high-level meaning (e.g., sushi is a type of Japanese food, a kettle is used for boiling water, a book is meant for reading, etc.).

Attribute is concerned with the object's specific features or characteristics (e.g., a ball is round, a handle is made of wood, etc.).

For the objects on the table in the image, your task is to create 17 instructions, which can either directly ask for an object or describe it using its spatial, semantic, or attribute properties (e.g., "pass me the item on the plate," "give me something that helps with drying hair," or "hand me the yellow object").

Each task is essentially a "pick" task, but the instruction should sound natural and realistic.

After giving the instruction, provide a more specific description that starts with "I need to pick up," and then clearly name the object, possibly with some additional spatial details to help locate it.

When describing a location, try to be as accurate as possible. Avoid using vague descriptions such as "in the middle/center of the table," "near," "beside," or "next to," as these could apply to many objects. Instead, use precise relative positioning, such as "to the left front of an object," "on top of an object," "between object A and object B," "to the right back of an object," or "behind an object."

When giving instructions, avoid mentioning the specific name of the object and instead use pronouns like "item," "object," or "device."

When providing attribute instructions, only list 1 or 2 properties of the object.

Your Tasks:
First, generating 2 tasks with direct references to the object name.
Then, generate 5 tasks **only** related to spatial properties (focusing on the location of the objects).
Next, generate 5 tasks related to semantic properties (focusing on the general meaning or purpose of the objects).
Finally, generate 5 tasks related to attribute properties (focusing on specific features of the objects).

For each task, follow this format:
**Start Task <task id>**
Instruction: ...
I need to pick up ...
**End Task <task id>**

Separate these 4 types of tasks by
### Tasks Related to Spatial Properties
### Tasks Related to Semantic Properties
### Tasks Related to Attribute Properties

Figure 17: **Prompt used to generate visual grounding task instructions and reasoning.**

In the given image, there is a table with several items placed on it in a messy manner.

Your task is to first imagine a long-horizon task based on the items in the image (such as organizing the table, making a sandwich, etc.). This task needs to be relatively long-term, meaning it should require about several steps to complete.

The second step is to provide a plan, where each step is a brief action description (e.g., Pick up sth and place it somewhere, Close sth, Open sth, Move sth to somewhere, etc.).

Output in the following format:
**Start Task**
Instruction: ...
1.
2.
…
N.
**End Task**

If you cannot think of an interesting task, simply output "Fail to think of a plan."

Note that the instruction and plan should be brief and precise.

Figure 18: **Prompt used to generate long-horizon task instructions and reasoning.**

# F  IMPLEMENTATION DETAILS

## F.1  ROBOT DATA INTERVALS

As mentioned in Sec. 3.2, we segment robot demonstrations into two types of intervals: *reasoning intervals* and *acting intervals*. Below, we detail what OneTwoVLA learns in each interval type.

1) *Reasoning intervals*, OneTwoVLA learns to:

- Predict [BOR] and the updated reasoning content $\hat{R}$ based on the latest reasoning content $R$.
- Predict [BOA] and actions based on the updated reasoning content $\hat{R}$.
- Predict actions based on the latest reasoning content $R$ without supervising [BOA]. This is to prevent incorrect action prediction if the model fails to update the reasoning promptly during deployment.

2) *Acting intervals*, OneTwoVLA learns to:

- Predict [BOA] and actions based on the latest reasoning content $R$.
- (Optional) Predict [BOR] based on outdated reasoning without supervising the reasoning content. This is included because we observe that during deployment, the model sometimes fails to enter the reasoning mode. Since predicting decision tokens is essentially a binary classification problem, and *acting intervals* are typically significantly longer than *reasoning intervals*, the model predominantly learns to predict [BOA], leading to an imbalanced classification problem. This optional training helps to increase the proportion of [BOR] predictions.

Additionally, it is important to note that *reasoning interval* during training is designed to encourage the model to learn the reasoning process more effectively. In real-world deployment, the robot only reasons at a small number of steps (rather than continuous intervals), ensuring that the overall operational efficiency is almost unaffected.

## F.2  POLICY TRAINING

As shown in Sec. 3.1, we use $\pi_0$ as our base model. For each task, we train the model for 30,000 steps on 8xH100 GPUs, requiring approximately 10 hours. We adopt training hyperparameters from $\pi_0$. We make two modifications to the original $\pi_0$'s input. Firstly, we use the current image $I_t$ and the reference image $I_{ref}$ as image observations. We incorporate $I_{ref}$ because the textual scene descriptions in reasoning may become outdated as the task progresses (e.g., an object's position described relative to the gripper becomes invalid upon gripper movement). Including $I_{ref}$, which corresponds to the image observation for the current reasoning content, helps prevent model confusion that might arise from potentially outdated textual descriptions. Second, we input not only the current robot proprioceptive states but also the proprioceptive states from 0.05 and 0.25 seconds earlier. This temporal context allows the model to generate more consistent and smooth actions during execution.

## F.3  DEPLOYMENT

In real-world deployment, we use the temporal ensemble (Zhao et al., 2023) technique to ensure smooth action execution. Specifically, in acting mode, the policy generates temporally overlapping action sequences every 0.2 seconds. At any given timestep, multiple predicted actions are averaged using exponential weighting to determine the actual executed actions.

Table 14 lists the computation time for $\pi_0$, along with the computation time for OneTwoVLA in acting mode for varying input token counts and in reasoning mode for varying output token counts, all of which are tested while processing two image inputs on an NVIDIA 4090 GPU. In acting mode, although OneTwoVLA has additional reasoning content as input and outputs an extra [BOA] compared to $\pi_0$, this has minimal impact on computation time and remains well below 0.2 seconds, thus execution efficiency is not affected in this mode. In reasoning mode, when the reasoning token count is low (less than 20 tokens), execution efficiency is unaffected; however, when reasoning content is lengthy (exceeding 100 tokens), the robot needs to pause for a few seconds. Nevertheless,

reasoning only occurs at a few critical moments, resulting in minimal impact on overall execution efficiency. For example, in one trial of the `Tomato-Egg` task, the entire long-horizon task takes 183 seconds, with reasoning occurring 5 times, totaling 16 seconds of reasoning time, which accounts for 8.7% of the total duration. Similarly, in one trial of the preparing `Mountain Fuji` task, the entire long-horizon task takes 135 seconds, with reasoning occurring 5 times, totaling 14 seconds of reasoning time, which accounts for 10.4% of the total duration.

| | # input tokens | # output tokens | computation time |
|---|---|---|---|
| $\pi_0$ | 20 | | 0.082 s |
| OneTwoVLA-Act-20 | 20 | 1 | 0.102 s |
| OneTwoVLA-Act-200 | 200 | 1 | 0.104 s |
| OneTwoVLA-Reason-20 | 200 | 20 | 0.853 s |
| OneTwoVLA-Reason-100 | 200 | 100 | 2.346 s |
| OneTwoVLA-Reason-200 | 200 | 200 | 4.361 s |

Table 14: **Computation times of $\pi_0$ and OneTwoVLA.** $\pi_0$'s input tokens consist solely of instruction $\ell$. OneTwoVLA's input tokens are typically longer, including instruction and latest reasoning content ($\ell$ and $R$). In acting mode (OneTwoVLA-Act rows), OneTwoVLA's output token is a single `[BOA]`. While in reasoning mode (OneTwoVLA-Reason rows), OneTwoVLA outputs `[BOR]` and updated reasoning content, $\hat{R}$. We showcase computation times when its output token length is 20, 100, and 200.

# G  OTHER FINDINGS

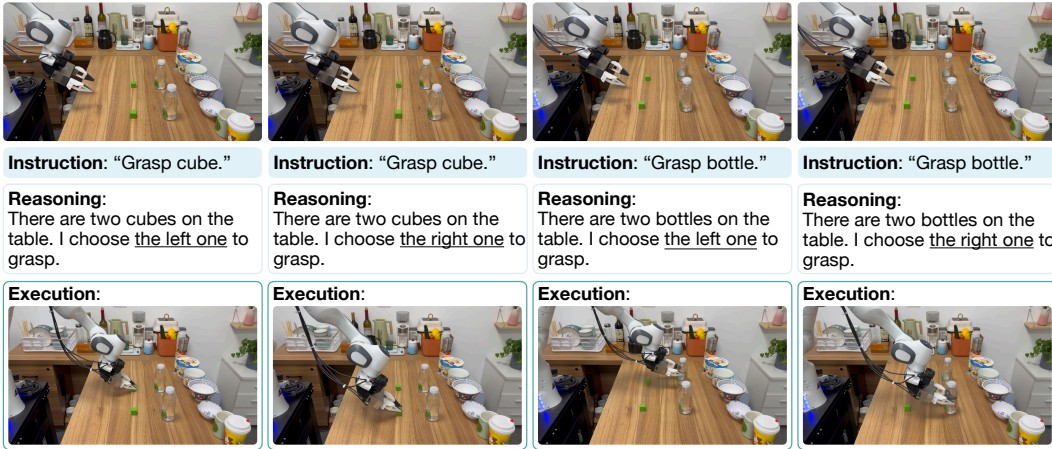

Figure 19: **Multi-modality task illustration.** Two cubes and two bottles are symmetrically placed on the table. When the instruction doesn't specify grasping the left or right object, OneTwoVLA can reason to grasp either the left or the right object, producing multi-modal actions.

## G.1  ONETWOVLA PRODUCES MULTI-MODAL ACTIONS

In this section, we design experiments to show OneTwoVLA's capability to produce multi-modal actions.

**Tasks and Evaluations.** Two identical cubes are symmetrically placed on a table, each with an identical bottle positioned symmetrically behind it. Using the UMI device, we collect 50 demonstrations for each of these four objects (totaling 200 demonstrations). Each demonstration instruction is either "Grasp the cube" or "Grasp the bottle," without specifying left or right. During testing, the object positions and the robotic gripper's initial pose remain fixed. Each method is tested 20 times per instruction.

**Comparative Methods.** 1) OneTwoVLA: For each demonstration, we explicitly include disambiguating reasoning content (e.g., specifying picking up the left or right object) to resolve the ambiguity. 2) $\pi_0$: The model receives the original instruction directly, without explicit disambiguation.

**Experimental Results.** As shown in Fig. 19, OneTwoVLA demonstrates multi-modal action capability by alternating between reasoning to grasp objects from either side. Specifically, in the "grasp cube" experiment, OneTwoVLA grasps the left cube 9 times and the right cube 11 times. In the "grasp bottle" experiment, it grasps the left bottle 8 times and the right bottle 12 times. OneTwoVLA achieves this balanced left-right performance because its reasoning process is probabilistic, which means the model can sample different decisions (such as whether to grasp from the left or right) based on predicted token probabilities, much like language models generate varied responses from the same input. In contrast, although flow matching (Lipman et al., 2022; Liu, 2022) or diffusion (Ho et al., 2020; Chi et al., 2023) algorithms theoretically enable multi-modality, $\pi_0$ consistently selects only the right-side objects, exhibiting only unimodal behavior, similar to observations in some other studies (Soare, 2024). Additionally, the disambiguating reasoning content helps the model fit actions more accurately. This is evidenced by $\pi_0$ occasionally failing to grasp the block, while OneTwoVLA consistently achieves precise grasps. Moreover, $\pi_0$'s action mean squared error (MSE) on the validation dataset is 56% higher than OneTwoVLA's. This interesting finding suggests that when training on large-scale, variable-quality robot datasets, detailed annotation of reasoning content may enhance action learning.

## G.2  ONETWOVLA PRODUCES REASONING-COMPLIANT ACTIONS

Our experiments show that the actions generated by OneTwoVLA consistently align with its reasoning, even when the reasoning itself is incorrect. This finding is similar to observations in previous

work (Zawalski et al., 2024). For example, in the `Hotpot` task, if OneTwoVLA occasionally reasons incorrectly about food locations, it proceeds to reach toward those incorrect positions. Similarly, in the `Open-World` experiment, OneTwoVLA moves to the object specified in its reasoning, even if that object does not align with the instruction. This indicates that OneTwoVLA's cognition and behavior are highly unified, showcasing synergistic reasoning and acting. Additionally, this interesting phenomenon may indicate that improving the model's reasoning ability (e.g., through additional vision-language data, using more powerful VLM as the base model, or more precise reasoning annotations) may contribute to generating more appropriate actions.

## H HARDWARE SETUP

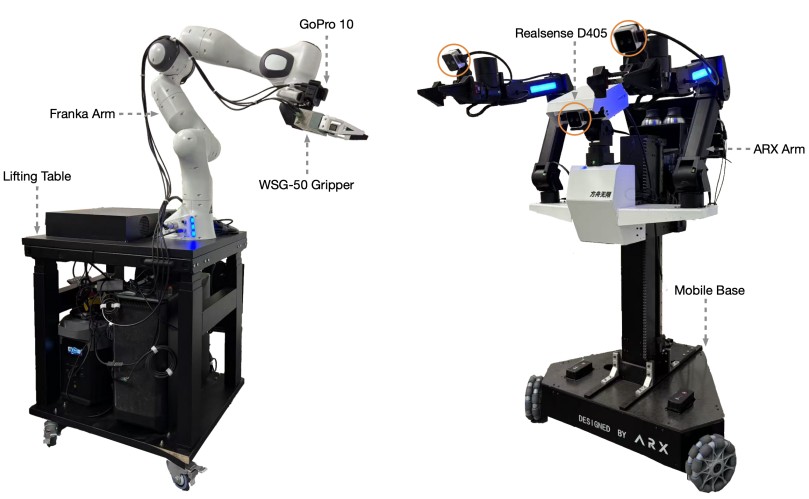

Figure 20: **Robot platform overview.** We employ two robot platforms: a single-arm Franka system (left) and a dual-arm ARX system (right).

We utilize two robot platforms. The primary platform (Fig. 20, left) is a single 7-DoF Franka arm equipped with a Weiss WSG-50 parallel-jaw gripper. A wrist-mounted GoPro camera with fisheye lens provides wide-angle observations. The arm is mounted on a custom height-adjustable table that can be pushed by a person—while not autonomous, this mobility allows us to evaluate the policy beyond traditional laboratory environments. The action space is 7-dimensional (6-DoF end-effector pose plus gripper width). Expert demonstrations for this platform are collected using UMI (Chi et al., 2024).

The second platform (Fig. 20, right) features two 6-DoF ARX arms with parallel-jaw grippers and a three-camera system (two wrist-mounted and one base-mounted). It also includes a holonomic wheeled base and a 1-DoF torso lift mechanism, though these components have not yet been utilized in our experiments. The resulting action space is 14-dimensional ($2 \times 7$). Expert demonstrations are collected via teleoperation using a Meta Quest headset.

# I FAILURE CASES

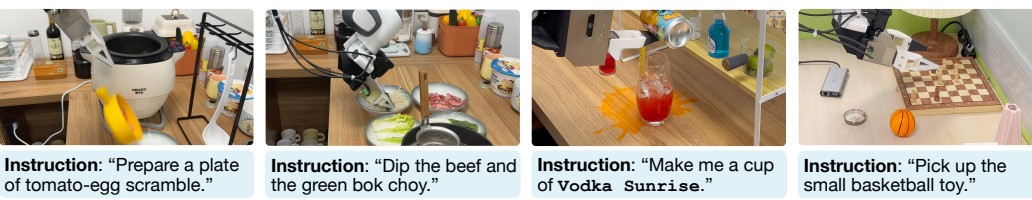

Figure 21: **Failure cases of OneTwoVLA.**

Despite the promising performance of OneTwoVLA, it still makes mistakes. Fig. 21 illustrates the main failure cases of OneTwoVLA. In the `Tomato-Egg` task, OneTwoVLA occasionally fails to grip the yellow plate containing tomato and egg liquid firmly enough, resulting in the plate being dropped (see the first column in Fig. 21). In the `Hotpot` task, OneTwoVLA sometimes misidentifies the location of the target ingredient. For instance, as shown in the Fig. 21 second column, the robot is instructed to pick up green bok choy but instead it attempts to pick up enoki mushrooms. The third column of Fig. 21 shows a case in `Cocktail` task, where OneTwoVLA fails to pour the orange juice accurately while preparing the `Vodka Sunrise`, causing the juice to spill. In the `Open-world` experiments, OneTwoVLA shows vulnerability when encountering objects that are not present in either the robot data or our synthesized vision-language data. For instance, as illustrated in the Fig. 21 fourth column, the robot consistently moves toward the chessboard despite being instructed to pick up the small basketball toy. We believe that training on larger robot datasets, as well as co-training with richer vision-language data, can further facilitate OneTwoVLA in learning fine-grained actions and improve generalization capabilities.

