# OpenReview forum: "OneTwoVLA: A Unified Vision-Language-Action Model with Adaptive Reasoning"
_ICLR.cc/2026/Conference — ICLR 2026 Poster_

### Official Review · Reviewer_2sCT · 2025-10-22

**Soundness:** 3
**Presentation:** 3
**Contribution:** 3
**Rating:** 6
**Confidence:** 4

**Summary:**

This paper proposes a single unified VLA model capable of both reasoning and acting, and adaptively switching between these two modes. It proposes a pipeline for synthesizing embodied reasoning-centric vision-language data to further enhance the model’s reasoning and generalization capabilities.

**Strengths:**

The paper is clearly written. The experimental design is appropriate and effectively demonstrates the proposed model’s capabilities. The model shows promising performance, particularly on long-horizon tasks, owing to the integration of reasoning and action within a unified framework.

**Weaknesses:**

Please refer to questions section.

**Questions:**

1. In Section 3.1, how is the adaptive inference capability learned? Is it acquired implicitly through training, or explicitly supervised with ground-truth signals? If it is the latter, how are such ground-truth labels defined and provided?

2. In Figure 2, the blue and green blocks are difficult to interpret. For instance, I understand the green part as representing System 1 and the blue as System 2. However, it is unclear why System 1 is further divided into instruction and reasoning inputs. How are these two types of inputs related to the two systems? The figure needs to be clarified.

3. The error detection capability is emphasized throughout the paper ( from the instruction to the conclusion) yet it is only qualitatively analyzed in Section 4.2, which is insufficient. If this ability is to be highlighted as a key contribution, it should be supported by quantitative evidence; otherwise, it would be better not to overemphasize it.

---

> ### Author Response · Authors · 2025-11-22
>
> > In Section 3.1, how is the adaptive inference capability learned? Is it acquired implicitly through training, or explicitly supervised with ground-truth signals? If it is the latter, how are such ground-truth labels defined and provided?
>
> This is indeed a key question, and we clarify the learning of the adaptive inference capability in Section 3.2. As you correctly inferred, it is explicitly supervised with ground-truth signals. Concretely, we first collect long-horizon demonstration trajectories. We then apply a fully automated pipeline that segments each trajectory into a sequence of intervals. There are two types of intervals: (1) *reasoning intervals*, which capture key steps that require model reasoning, and (2) *acting intervals*, where the model primarily learns to predict actions. During training, we supervise the decision tokens according to the interval type and the freshness of reasoning content $R$. In reasoning intervals, the ground-truth decision token is [BOR] if the current reasoning $R$ is stale (i.e., needs updating); once $R$ has been updated, the ground truth becomes [BOA]. In acting intervals, the model always learns to predict [BOA]. We provide a more detailed explanation of this supervision scheme in Appendix F.1.
>
> > In Figure 2, the blue and green blocks are difficult to interpret. For instance, I understand the green part as representing System 1 and the blue as System 2. However, it is unclear why System 1 is further divided into instruction and reasoning inputs. How are these two types of inputs related to the two systems? The figure needs to be clarified.
>
> Thank you for pointing this out. We believe you are referring to Figure 3 rather than Figure 2. We agree that the original figure was not sufficiently clear.
>
> The reason System 1 is visually divided into “instruction” and “reasoning” inputs is that, in the acting mode (System 1), the model conditions on the latest reasoning content to produce actions. This reasoning content includes a scene description, a high-level plan, a historical summary, and the immediate next step for the robot to execute. Formally, the acting mode can be written as $A_t\sim\pi_{\theta}(\cdot | I_{t}^{1:n}, I_{\text{ref}}^{1:n}, \ell, R, s_t)$, where $\ell$ is the language instruction and $R$ is the latest reasoning content.
>
> In the reasoning mode (System 2), the model also conditions on the previous reasoning content and outputs updated reasoning: $\hat{R}\sim\pi_{\theta}(\cdot | I_{t}^{1:n}, I_{\text{ref}}^{1:n}, \ell, R)$.
>
> Thus, in both acting mode (System 1) and reasoning mode (System 2), the model input includes both **the language instruction and the reasoning content**. To address the ambiguity, we have added a new, clearer diagram in the appendix, which separates the two modes into distinct subfigures and annotates them with the corresponding mathematical notation, making the roles of the inputs more explicit.
>
> > The error detection capability is emphasized throughout the paper ( from the instruction to the conclusion) yet it is only qualitatively analyzed in Section 4.2, which is insufficient. If this ability is to be highlighted as a key contribution, it should be supported by quantitative evidence; otherwise, it would be better not to overemphasize it.
>
> Thank you for this thoughtful suggestion. We agree that error detection and recovery is a critical capability. In addition to the qualitative examples in Section 4.2, our original submission already includes a detailed **quantitative** comparison in Table 1 (lines 361–368). Across two tasks, there are 10 error events in total. Our method, OneTwoVLA, successfully recovers from 8 out of these 10 errors, achieving an 80% success rate in error detection and recovery. In contrast, the baselines $\pi_0$ and Dual-System achieve success rates of only 57.1% and 58.3%, respectively. We will revise the main text to more clearly highlight this quantitative evidence when discussing error detection and recovery, so that the emphasis in the narrative is better aligned with the presented empirical results.

---

### Official Review · Reviewer_yjpV · 2025-10-29

**Soundness:** 3
**Presentation:** 3
**Contribution:** 3
**Rating:** 6
**Confidence:** 5

**Summary:**

OneTwoVLA introduces a VLA framework that allows a single model to adaptively decide when to reason and when to act, bridging high-level reasoning and low-level control. It employs special decision tokens ([BOR]/[BOA]) to autonomously switch between reasoning and acting modes. Experiments on long-horizon real-world manipulation tasks demonstrate that OneTwoVLA achieves superior success rates.

**Strengths:**

S1. It simultaneously equips the VLA model with reasoning capability and action prediction ability, which is important for robotics.

S2. The authors conducted a large number of experiments, including "out-of-lab" real-world tests. It is amazing.

S3. The writing is very thorough and careful, with a large appendix experiments and visualizations included.

**Weaknesses:**

W1. Lack of key ablations. The paper lacks experiments that use different reasoning components as conditions, e.g., using only “plan” or “Now I need to …”. Not every lab has the ability to collect all key-step annotations, and I also do not believe every reasoning element contributes positively to all tasks. For instance, in dynamic environments, is the “Historical Summary” really helpful? And in complex scenes, could detailed “scene descriptions” actually introduce unnecessary redundancy?

W2. Impact on generalization. Would using different subsets of reasoning outputs as conditions affect the model’s generalization ability?

W3. Table 14 only compares inference time across reasoning lengths. Did the authors investigate how different reasoning lengths affect manipulation accuracy? Also, what reasoning token length was used in the main experiments?

W4. What is the additional time cost introduced to the execution head by different reasoning token lengths?

W5. If the number of condition tokens increases, it may significantly slow down the DDIM output efficiency. In addition, could the authors provide a real-robot control frequency comparison experiment to support this?

W6. Since the authors already built an automated data labeling pipeline, could it be extended to modify pretraining datasets such as OXE? This could enable the model to develop reasoning-and-action collaboration abilities across broader domains, not just in the downstream tasks.

**Questions:**

Q1. In the out-of-lab experiments, under such complex backgrounds, especially when using a GoPro with a wide field of view, is it truly possible to achieve stable manipulation? Is this manipulation capability mainly derived from the pretrained knowledge of the base model?

Q2. The appendix layout could be improved, it contains excessive blank space and is overly long, making it difficult for reviewers to find key information efficiently.

Q3. It would be helpful if the authors could add a discussion comparing with Pi_0.5 and ThinkAct. Of course, a quantitative comparison is not strictly necessary.

Q4. This paper presents a very good idea, but it lacks detailed exploration and validation. I will adjust my score based on the authors’ response.

---

> ### Author Response · Authors · 2025-11-22
>
> Thank you so much for your review and suggestions. Below are our respones.
>
> > W1. Lack of key ablations. The paper lacks experiments that use different reasoning components as conditions, e.g., using only “plan” or “Now I need to …”. Not every lab has the ability to collect all key-step annotations, and I also do not believe every reasoning element contributes positively to all tasks. For instance, in dynamic environments, is the “Historical Summary” really helpful? And in complex scenes, could detailed “scene descriptions” actually introduce unnecessary redundancy?
>
> **A1**: Thank you for the suggestion. We added new ablation studies on the Hotpot task. The original reasoning content includes four fields: “Scene Description,” “Plan,” “Historical Summary,” and “Now I need to …”. We trained OneTwoVLA with four reduced reasoning configurations: (1) a single “Plan” field (Only-Plan), (2) a single “Now I need to …” field (Only-Next-Step), (3) our full method without “Historical Summary” (No-Historical-Summary), and (4) our full method without “Scene Description” (No-Scene-Description). We evaluated the original OneTwoVLA and these variants over 10 trials.
>
> **Table 1**. Task success rates on Hotpot for different reasoning configurations (successes/trials).
> | Reasoning configuration |Scene description|Plan|Historical summary|Now I need to| Task success rate |
> |---|---|---|---|---|---|
> |OneTwoVLA (ours) |✓|✓|✓|✓| 9/10|
> |Only-Plan |✓|✕|✕|✕| 5/10 |
> |Only-Next-Step |✕|✕|✕|✓|7/10|
> |No-Historical-Summary |✓|✓|✕|✓|8/10|
> |No-Scene-Description|✕|✓|✓|✓|6/10|
>
> Omitting reasoning components affects performance to varying degrees. On Hotpot, progress-tracking components—“Historical Summary” and “Now I need to …”—are important. The Only-Plan variant, which lacks these fields, often repeatedly picks up the beef. Explicitly outlining the next step helps the model grasp the instructed vegetable. The Only-Plan configuration exhibits the highest rate of grasping the wrong vegetable. The “Scene Description” also helps locate the target and improves grasp accuracy; Only-Next-Step, which lacks this field, moves toward the correct area but closes the gripper outside the vegetable plate and ultimately fails. Interestingly, removing “Historical Summary” results in only a minor performance drop, possibly because it overlaps with the “Now I need to …” field for this task.
>
> We also evaluated human–robot interaction on Hotpot, where the human requests adding a vegetable and the robot must ask which kind.
>
> **Table 2**. Human–robot interaction performance on Hotpot (successes/interactions).
> | Reasoning configuration |Scene description|Plan|Historical summary|Now I need to| # successes / # interactions |
> |---|---|---|---|---|---|
> |OneTwoVLA (ours) |✓|✓|✓|✓| 10/10|
> |Only-Plan|✓|✕|✕|✕| 0/10 |
> |Only-Next-Step|✕|✕|✕|✓|10/10|
> |No-Historical-Summary|✓|✓|✕|✓|10/10|
> |No-Scene-Description|✕|✓|✓|✓|10/10|
>
> The Only-Plan variant fails to respond to human queries, whereas the other reduced configurations match OneTwoVLA’s interaction capability.
>
> We further tested error detection and recovery.
>
> **Table 3**. Error detection and recovery on Hotpot (successful recoveries/error occurrences).
> | Reasoning configuration |Scene description|Plan|Historical summary|Now I need to| # successes recoveries / # error occurrences |
> |---|---|---|---|---|---|
> |OneTwoVLA (ours) |✓|✓|✓|✓|4/4|
> |Only-Plan|✓|✕|✕|✕| 0/6 |
> |Only-Next-Step|✕|✕|✕|✓|3/4|
> |No-Historical-Summary|✓|✓|✕|✓|5/5|
> |No-Scene-Description|✕|✓|✓|✓|0/5|
>
> Notably, Only-Plan and No-Scene-Description fail to detect errors. Only-Next-Step shows a slight drop, likely due to the absence of the “Scene Description” field, which explicitly describes the error state (e.g., the gripper is closed but no strainer is inside).
>
> In summary for Hotpot: (1) using Only-Plan without progress tracking leads to a substantial performance drop; (2) Only-Next-Step and (3) No-Scene-Description are generally acceptable but slightly affect grasp accuracy and error detection; (4) removing “Historical Summary” has the smallest impact.
>
> However, different tasks may prefer different subsets of reasoning components. For example, the “Historical Summary” field—least important for Hotpot—can be crucial for repeated steps. A plan like “Add 5 spoonfuls of sugar” requires this field to record how much sugar has been added. Overall, our adaptive reasoning paradigm supports multiple reasoning configurations, allowing room to tailor components to specific tasks.

---

> > ### Author Response · Authors · 2025-11-22
> >
> > > W2: Impact on generalization. Would using different subsets of reasoning outputs as conditions affect the model’s generalization ability?
> >
> > **A2**: Yes. As shown by the ablations in **A1**, the chosen subset of reasoning components affects generalization under train–test distribution shifts (e.g., plate placement.).
> >
> > For generalization to unseen tasks and instructions, the critical signal in visual grounding is the “Now I need to…” field, which explicitly names the object referenced in the instruction. Without this field (see the π0 row in Table 3), the policy achieves only a 3% success rate. Similarly, in the generalizable planning setting—which relies on the “Plan” field—removing this field reduces the success rate to 6%.
> >
> > > W3. Table 14 only compares inference time across reasoning lengths. Did the authors investigate how different reasoning lengths affect manipulation accuracy? Also, what reasoning token length was used in the main experiments?
> >
> > **A3**: In our format,  reasoning length is task-dependent: tasks with more steps require longer reasoning. It is not a tunable hyperparameter. We did not treat length as an independent variable and therefore did not directly study its effect on manipulation accuracy. In the main experiments, reasoning token length varied by task. The table below reports the maximum observed reasoning lengths for each task.
> >
> > | Task | Maximum reasoning length (tokens) |
> > |---|---|
> > | Hotpot | 386 |
> > | Tomato–Egg | 350 |
> > | Cocktail | 405 |
> >
> > > W4. What is the additional time cost introduced to the execution head by different reasoning token lengths?
> >
> > **A4**: Longer reasoning sequences add minimal overhead. We quantify execution overhead across reasoning lengths using dummy inputs. As shown by the OneTwoVLA-Act-20 and OneTwoVLA-Act-200 entries in Table 14, increasing the reasoning token length from 20 to 200 adds about 2 ms to inference time (from 102 to 104 ms).
> >
> > > W5. If the number of condition tokens increases, it may significantly slow down the DDIM output efficiency. In addition, could the authors provide a real-robot control frequency comparison experiment to support this?
> >
> > **A5**:  As noted in **A4**, longer reasoning sequences add little latency to action inference. Prefill and diffusion denoising process tokens in parallel, and KV caching avoids re-forwarding condition tokens at each diffusion step. Consequently, OneTwoVLA operates at the same control frequency (5 Hz) as the non-reasoning baseline π0.
> >
> > > W6. Since the authors already built an automated data labeling pipeline, could it be extended to modify pretraining datasets such as OXE? This could enable the model to develop reasoning-and-action collaboration abilities across broader domains, not just in the downstream tasks.
> >
> > **A6**: Thank you for the suggestion. This is a promising direction but would require substantial compute. We are actively investigating it in an ongoing project.

---

> > > ### Author Response · Authors · 2025-11-22
> > >
> > > > Q1. In the out-of-lab experiments, under such complex backgrounds, especially when using a GoPro with a wide field of view, is it truly possible to achieve stable manipulation? Is this manipulation capability mainly derived from the pretrained knowledge of the base model?
> > >
> > > **A1**:  Stable out-of-lab manipulation is primarily enabled by (1) UMI with the Franka hardware setup and (2) diverse training data. UMI’s deployment stack applies action interpolation and asynchronous layers for policy inference and low-level robot control, yielding safe, smooth actions. We also collect robot data across multiple environments (e.g., 16 in-the-wild environments for open-world visual grounding). Strong in-the-wild manipulation capability driven by diverse training data is widely reported [1–4].
> > >
> > > Pretrained knowledge from the foundation model also helps, especially for out-of-distribution performance. Prior work shows that pretrained foundation models generalize better than from-scratch baselines [5–7].
> > >
> > > [1] Cheng Chi, Zhenjia Xu, Chuer Pan, Eric Cousineau, Benjamin Burchfiel, Siyuan Feng, Russ Tedrake, and Shuran Song. Universal manipulation interface: In-the-wild robot teaching without in-the-wild robots. arXiv:2402.10329, 2024.
> > >
> > > [2] Physical Intelligence, Kevin Black, Noah Brown, James Darpinian, Karan Dhabalia, Danny Driess, Adnan Esmail, Michael Equi, Chelsea Finn, Niccolo Fusai, et al. π0.5: A vision-language-action model with open-world generalization. arXiv:2504.16054, 2025.
> > >
> > > [3] Fanqi Lin, Yingdong Hu, Pingyue Sheng, Chuan Wen, Jiacheng You, and Yang Gao. Data scaling laws in imitation learning for robotic manipulation. arXiv:2410.18647, 2024.
> > >
> > > [4] https://www.sunday.ai/
> > >
> > > [5] Burns, K., Witzel, Z., Hamid, J. I., Yu, T., Finn, C., and Hausman, K. What makes pre-trained visual representations successful for robust manipulation? arXiv:2312.12444, 2023.
> > >
> > > [6] Zitkovich, B., Yu, T., Xu, S., Xu, P., Xiao, T., Xia, F., ... and Han, K. RT-2: Vision-language-action models transfer web knowledge to robotic control. In Conference on Robot Learning (pp. 2165–2183). PMLR, 2023.
> > >
> > > [7] Kim, M. J., Pertsch, K., Karamcheti, S., Xiao, T., Balakrishna, A., Nair, S., ... and Finn, C. OpenVLA: An open-source vision-language-action model. arXiv:2406.09246, 2024.
> > >
> > > > Q2. The appendix layout could be improved, it contains excessive blank space and is overly long, making it difficult for reviewers to find key information efficiently.
> > >
> > > **A2**: Thank you for the feedback. We will streamline the appendix in the revision by reducing blank space, consolidating redundant content, and adding navigation aids.
> > >
> > > > Q3. It would be helpful if the authors could add a discussion comparing with Pi_0.5 and ThinkAct. Of course, a quantitative comparison is not strictly necessary.
> > >
> > > **A3**: Our main difference from π0.5 is adaptive reasoning. π0.5 uses an always-reason strategy, performing reasoning before every action; consequently, its reasoning focuses on immediate subtasks. In contrast, our method reasons adaptively at critical moments, deciding when to reason and when to act. This flexibility enables more detailed reasoning without reducing inference frequency.
> > >
> > > ThinkAct is a dual-system method with separately trained models, whereas OneTwoVLA is unified. Compared with the dual-system baseline in our study, ThinkAct’s high-level system is specifically trained for embodied reasoning, and the two systems communicate via a visual-plan latent vector, which may mitigate limited mutual understanding. However, ThinkAct reasons at a fixed frequency rather than adaptively, so high-level latency may still affect reactivity. Moreover, its manipulation performance is under-evaluated so far, with results primarily in simulation.
> > >
> > > > Q4. This paper presents a very good idea, but it lacks detailed exploration and validation. I will adjust my score based on the authors’ response.
> > >
> > > **A4**: Thank you for your feedback and review. We have added additional results and explanations in our response and will continue to revise the manuscript. We hope our response addresses your questions, and we look forward to further discussion.

---

### Official Review · Reviewer_uSyR · 2025-10-30

**Soundness:** 2
**Presentation:** 4
**Contribution:** 3
**Rating:** 6
**Confidence:** 4

**Summary:**

This paper presents OneTwoVLA, a unified vision-language-action model capable of both reasoning (System Two) and acting (System One) within a single framework. Unlike prior dual-system approaches that separate high-level reasoning from low-level control, OneTwoVLA adaptively switches between reasoning and action—engaging explicit language reasoning at critical moments while efficiently generating actions otherwise. The authors further propose a scalable pipeline for synthesizing embodied, reasoning-centric vision-language data to co-train with real robot data. Extensive experiments demonstrate that OneTwoVLA significantly outperforms existing methods across four key dimensions: long-horizon task planning, error detection and recovery, natural human-robot interaction, and generalizable visual grounding, highlighting the promise of unified models for embodied intelligence.

**Strengths:**

OneTwoVLA’s main strengths lie in its elegant unification of reasoning and action within a single adaptive model, which effectively eliminates the latency and coordination issues common in dual-system designs. It also demonstrates impressive generalization by leveraging a scalable pipeline that synthesizes embodied reasoning-centric vision-language data, significantly enhancing performance on unseen tasks. Moreover, the model enables natural and context-aware human-robot interaction, showing a level of interpretability and adaptability in current robotic systems.

One more thing:  The authors provide highly detailed supplementary content, including rich qualitative examples, visualizations, and additional experiments, which greatly improve the clarity and reproducibility of the work.

**Weaknesses:**

1. The reasoning-centric synthetic dataset appears essential to the model’s overall performance, but it remains unclear how much it directly contributes to robotic manipulation capabilities. An explicit ablation study comparing models trained with and without this reasoning data — or analyses showing how it benefits manipulation compared to standard vision-language data — would make the claim far more convincing.

2. The model’s generalization mainly lies in reasoning, while action-level generalization remains limited. Given that large language models already generalize well in reasoning for simple robotic tasks, it would be more meaningful to explore whether OneTwoVLA can generalize in low-level manipulation, such as grasping unseen objects.

**Questions:**

1. Is the sequential dual-phase design (reasoning first, then acting) fundamentally necessary? Have the authors considered a mechanism that allows simultaneous reasoning and acting, similar to how humans can reason while performing actions, rather than pausing execution each time reasoning is invoked? It would be valuable for the authors to discuss how such an asynchronous or parallel architecture might be designed and what challenges it would introduce.

2. Regarding the synthetic data filtering process, line 1536 mentions that incorrect images are excluded from evaluation. Could the authors clarify how this filtering was conducted — was it done manually, automatically, or through a hybrid approach? More detail on the evaluation pipeline and its reproducibility would improve transparency.

If the authors can address these concerns or provide more evidence supporting their design choices, I would consider raising my score.

---

> ### Author Response · Authors · 2025-11-22
>
> > The reasoning-centric synthetic dataset appears essential to the model’s overall performance, but it remains unclear how much it directly contributes to robotic manipulation capabilities. An explicit ablation study comparing models trained with and without this reasoning data — or analyses showing how it benefits manipulation compared to standard vision-language data — would make the claim far more convincing.
>
> We thank the reviewer for highlighting the need for this clarification. Table 4 compares models trained with our embodied reasoning-centric VL data (OneTwoVLA-VL) versus without it (OneTwoVLA, and π0). The experiments demonstrate that our synthetic reasoning-centric vision-language data enhances the model's generalizable visual grounding capabilities, enabling it to correctly handle objects unseen in the robot data. While many works [1, 2] show that standard vision-language data (e.g., public VQA, detection datasets) can also improve generalization, the advantage of our synthesized dataset is that it allows us to **customize** the questions and answers to directly inject the capabilities we want embodied models to possess, such as planning abilities, understanding of spatial/attribute/semantic properties in visual grounding, and recognition of unseen objects. In contrast, public datasets are often noisy and difficult to filter for such specific embodied reasoning knowledge, making our targeted synthetic approach significantly more sample-efficient.
>
> > The model’s generalization mainly lies in reasoning, while action-level generalization remains limited. Given that large language models already generalize well in reasoning for simple robotic tasks, it would be more meaningful to explore whether OneTwoVLA can generalize in low-level manipulation, such as grasping unseen objects.
>
> This is an excellent point. Our experiments indeed show that co-training with our synthetic vision-language data enables the model to correctly handle objects unseen in the robot data. We acknowledge that the primary focus of our visual grounding experiments was to validate the model's ability to recognize and identify these objects correctly, rather than to test its low-level manipulation dexterity on them. We posit that generalization in high-level reasoning and low-level action are distinct challenges that benefit from different data sources. The ability to reason about and identify unseen objects (the "what" and "where") can be effectively enhanced through co-training with diverse vision-language data, as we have shown. However, we believe that achieving generalization in fine-grained, low-level manipulation—such as grasping objects with novel or complex geometries, or executing entirely new motions and skills (the "how")—is a problem that must be addressed with **rich and diverse robot behavior data**.
>
> [1] π0.5: a Vision-Language-Action Model with Open-World Generalization
>
> [2] Vlaser: Vision-Language-Action Model with Synergistic Embodied Reasoning

---

> > ### Author Response · Authors · 2025-11-22
> >
> > > Is the sequential dual-phase design (reasoning first, then acting) fundamentally necessary? Have the authors considered a mechanism that allows simultaneous reasoning and acting, similar to how humans can reason while performing actions, rather than pausing execution each time reasoning is invoked? It would be valuable for the authors to discuss how such an asynchronous or parallel architecture might be designed and what challenges it would introduce.
> >
> > We thank the reviewer for this insightful question regarding a more integrated architecture. The current sequential design is a deliberate choice for robustness, but we agree that an asynchronous model that allows for simultaneous reasoning and acting is a compelling direction for future work.
> >
> > One potential design for such a parallel architecture would be to have the model generate the reasoning content token-by-token while, concurrently, the action expert predicts a continuous stream of actions conditioned on the most up-to-date, partially-generated reasoning. However, this introduces significant challenges:
> >
> > (1) **Acting on Incomplete Information**: The model would be forced to act based on an incomplete and evolving thought process. Critical information or a change in strategy often appears at the end of a reasoning sequence (e.g., "I should pick up the cup, but first, I need to move the obstacle"). Acting prematurely on the initial part of the plan ("I should pick up the cup") could lead to incorrect or suboptimal actions.
> >
> > (2) **Risk of Irrecoverable Errors**: A premature action based on partial reasoning might lead the robot into a state from which recovery is difficult or impossible. For instance, the model might start pouring an ingredient before it has finished reasoning about the correct target container, causing a spill.
> >
> > Given these challenges, our current adaptive-but-sequential framework represents a robust and practical choice, ensuring that actions are guided by complete and coherent reasoning at critical junctures, thereby minimizing errors.
> >
> > > Regarding the synthetic data filtering process, line 1536 mentions that incorrect images are excluded from evaluation. Could the authors clarify how this filtering was conducted — was it done manually, automatically, or through a hybrid approach? More detail on the evaluation pipeline and its reproducibility would improve transparency.
> >
> > To clarify, the exclusion applies only to the data-quality examination. Samples with incorrect images are counted only in the “Wrong Image” column in Table 11; we do not evaluate their accompanying text. For training, we use all synthetic data without filtering. We apologize for the ambiguity and will clarify this in the revision.

---

> > > ### Comment · Reviewer_uSyR · 2025-11-26
> > > **Thanks for author rebuttal**
> > >
> > > Thank you for addressing my concerns. While your responses have clarified some points, I still find the core contributions—specifically the unified VLA structure and reasoning dataset—somewhat limited in their impact.
> > >
> > > Given the clarifications, I believe the paper can be accepted, but I would support reconsideration if other reviewers or the AC have more reasons for rejection. Therefore, I will maintain my current score.
> > >
> > > Thank you again for your work.

---

### Official Review · Reviewer_EYxJ · 2025-11-02

**Soundness:** 2
**Presentation:** 2
**Contribution:** 2
**Rating:** 4
**Confidence:** 3

**Summary:**

- This paper introduces OneTwoVLA, a vision-language-action model that can perform both acting and reasoning.

- OneTwoVLA adaptively reasons at critical moments during execution (e.g., upon completing subtasks, detecting errors, or requiring human inputs).

- This paper also designs a pipeline for synthesizing embodied reasoning-centric vision-language data, used for co-training with
robot data.

**Strengths:**

- Improving the reasoning capabilities of current embodied systems is crucial, and enabling adaptive reasoning is particularly insightful.

- OneTwoVLA is highly practical, but it would be better if the authors could validate its generalization ability on public benchmarks.

**Weaknesses:**

- OneTwoVLA inherits the model architecture of pi0 (VLM for reasoning and flow-matching policy for acting). Why is it described as a single unified vision-language-action model?

- Regarding the issue raised by the authors that "System Two may produce intermediate contents that System One cannot execute," how does OneTwoVLA address this problem? Why does the standard two-system VLA suffer from limited mutual understanding, while OneTwoVLA does not?

- In my view, compared to previous works, OneTwoVLA does not introduce architectural innovations; essentially, it still relies on a VLM for reasoning and subtask generation, with an action expert responsible for execution. The main difference lies in the fact that the authors heuristically designed certain scenarios requiring re-planning and annotated corresponding data to fine-tune pi0.

- Currently, there are many works focused on embodied reasoning. Could the authors evaluate OneTwoVLA on some public benchmarks to demonstrate its multimodal reasoning and planning capabilities? For example, the following works:

[1] RoboBrain 2.0 Technical Report

[2] Embodied Arena: A Comprehensive, Unified, and Evolving Evaluation Platform for Embodied AI

- Could the authors report the reasoning time overhead introduced by adaptive reasoning? For the baseline, is it possible to improve performance by increasing the frequency of System 2 execution?

**Questions:**

Please see the Weaknesses section.

---

> ### Author Response · Authors · 2025-11-22
>
> > OneTwoVLA inherits the model architecture of pi0 (VLM for reasoning and flow-matching policy for acting). Why is it described as a single unified vision-language-action model?
>
> Thank you for this important question. We emphasize "single unified" in contrast to dual-system approaches, such as Hi Robot [1], which contain a separate System Two for high-level reasoning and a System One that translates intermediate contents into low-level actions—typically trained separately. We consider VLA models comprised of a VLM backbone and an action head to be single unified models following a mixture-of-experts paradigm. The "single unified" claim therefore emphasizes our use of a single VLA model capable of both reasoning and acting, and critically, adaptively switching between these two modes, highlighting this paradigm rather than inventing new network modules. While we build upon the π₀ architecture, our core contribution is the adaptive reasoning framework itself, which is designed to be general and applicable to most existing VLAs. We selected π₀ as the base model due to its demonstrated strong performance.
>
> [1] Hi Robot: Open-Ended Instruction Following with Hierarchical Vision-Language-Action Models
>
> > Regarding the issue raised by the authors that "System Two may produce intermediate contents that System One cannot execute," how does OneTwoVLA address this problem? Why does the standard two-system VLA suffer from limited mutual understanding, while OneTwoVLA does not?
>
> We appreciate the opportunity to clarify how OneTwoVLA resolves the mutual understanding issue. OneTwoVLA mitigates this problem through its unified architecture and co-training paradigm. By training a single model on a rich mixture of robot data and vision-language data, OneTwoVLA develops a synergistic and consistent understanding of both reasoning and acting: the reasoning is grounded in what is visually perceived and physically achievable by the robot, while the actions are directly guided by this grounded reasoning. In dual-system frameworks, the lack of mutual understanding arises because the two systems are trained separately, often on different data distributions (e.g., System Two on vision-language data, System One on robot action data). As we demonstrated in our experiments, in the Tomato-Egg task, the dual-system baseline's System Two (Gemini Pro 2.5), trained on broader vision-language datasets, commands adding green onion when none is present in the scene. While this instruction appears reasonable from a culinary perspective, System One cannot execute it as it has not been trained on such data. OneTwoVLA avoids this issue because its reasoning and perception are intrinsically linked within the same model.
>
> > In my view, compared to previous works, OneTwoVLA does not introduce architectural innovations; essentially, it still relies on a VLM for reasoning and subtask generation, with an action expert responsible for execution. The main difference lies in the fact that the authors heuristically designed certain scenarios requiring re-planning and annotated corresponding data to fine-tune pi0.
>
> Thank you for your perspective on the contributions of our work. Our contribution is not model architectural innovation, but rather the adaptive reasoning paradigm that emphasizes synergistic and adaptive reasoning-acting capabilities. Crucially, this paradigm is general and can be readily adapted to most VLAs with minimal modifications, including those composed of VLM and action expert (e.g., π₀), as well as auto-regressive architectures (e.g., FAST). Our experiments validate the significant advantages of OneTwoVLA across multiple capabilities, including long-horizon planning, error recovery, human-robot interaction, and visual grounding.
>
> > Currently, there are many works focused on embodied reasoning. Could the authors evaluate OneTwoVLA on some public benchmarks to demonstrate its multimodal reasoning and planning capabilities? For example, the following works: [1] RoboBrain 2.0 Technical Report [2] Embodied Arena: A Comprehensive, Unified, and Evolving Evaluation Platform for Embodied AI
>
> We are grateful for the suggestion to evaluate on public embodied reasoning benchmarks. The benchmarks mentioned are excellent for evaluating the embodied reasoning capabilities of foundation VLMs on vision-language tasks (e.g., visual question answering, detection, etc.). However, our focus is not on developing powerful foundation VLMs, but rather on how to effectively **post-train** a pretrained foundation model into a robot manipulation policy with tightly coupled reasoning and acting. Therefore, our evaluation is centered on physical robotics tasks rather than VLM-only benchmarks.

---

> > ### Author Response · Authors · 2025-11-22
> >
> > > Could the authors report the reasoning time overhead introduced by adaptive reasoning? For the baseline, is it possible to improve performance by increasing the frequency of System 2 execution?
> >
> > We thank the reviewer for asking about the reasoning time overhead. We report the reasoning time in Figure 2 for the Tomato-Egg task: reasoning takes 16 seconds, accounting for 8.7% of the total execution time of 184 seconds. In the dual-system approach, System Two reasons at fixed intervals (every 5 seconds). Since we use Gemini 2.5 Pro, given the significant inference latency (average 4.6 seconds) of such a large model, this frequency is already near the maximum practical limit for maintaining task flow. Therefore, increasing frequency cannot resolve the dual-system's latency issues, nor can it address the lack of mutual understanding between the two systems.

---

### Meta-Review · Area_Chair_C1sQ · 2026-01-10

**Summary:**

The paper introduces OneTwoVLA, a unified vision-language-action model that adaptively switches between explicit reasoning and action generation. While the reviewers generally appreciated the extensive real-world experiments and the practical benefits of the adaptive framework, several key concerns were raised:

+ Architectural Novelty: Multiple reviewers (EYxJ, uSyR) pointed out that the model heavily inherits the $\Pi_0$ architecture, leading to questions about the significance of the technical innovation beyond heuristic data labeling.

+ Impact of Reasoning on Manipulation: Reviewer uSyR questioned the direct contribution of the synthetic reasoning dataset to low-level manipulation dexterity versus high-level planning.

+ Ablation and Generalization: Reviewer yjpV highlighted a lack of initial evidence regarding which specific reasoning fields (e.g., "Historical Summary" vs. "Plan") truly drive performance and how they affect generalization to unseen tasks.

+ Mechanism Clarity: Reviewers requested clarification on how the adaptive switching (via [BOR]/[BOA] tokens) is supervised and whether a parallel reasoning-acting architecture would be more efficient.

The authors' rebuttal addressed many of these points by providing new ablation studies on reasoning configurations, clarifying the supervision scheme, and demonstrating improved success rates on long-horizon tasks compared to dual-system baselines. While some reviewers remained skeptical about the overall impact of the contributions, the consensus leaned toward a marginal acceptance based on the empirical strength of the work.

**Reviewer Concerns:**

Addressed Concerns:

+ Technical Supervision and Implementation: The authors clarified the explicit supervision scheme for the adaptive [BOR]/[BOA] tokens using ground-truth signals from segmented trajectories. They also addressed confusion regarding the system's inference flow and input types.

+ Ablation Studies: In response to requests for deeper analysis, the authors provided new ablation results on the Hotpot task, quantifying the relative importance of "Scene Description," "Plan," and "Historical Summary" for task success, error recovery, and human interaction.

+ Inference Efficiency: The rebuttal provided concrete data on reasoning time (accounting for ~8.7–10.4% of total execution) and confirmed that action inference overhead remains minimal (approx. 2ms) despite longer reasoning prefixes.

+ Visual Grounding Generalization: The authors demonstrated that co-training with synthetic data significantly improves the model's ability to handle objects unseen in the robot training data.

Outstanding Concerns:

+ Architectural Novelty: Reviewers EYxJ and uSyR remained skeptical about the fundamental technical innovation, noting that the model's core architecture heavily relies on $\Pi_0$ and that the primary contribution lies in heuristic data annotation rather than new network modules.

+ Low-Level Manipulation Generalization: While high-level reasoning generalizes well, reviewers pointed out that the model's ability to generalize to novel low-level motor skills (the "how" of manipulation) remains limited and was not the primary focus of the evaluation.

+ Standardized Benchmarking: The authors declined to evaluate the model on public embodied reasoning benchmarks (e.g., RoboBrain 2.0), choosing instead to focus exclusively on their internal physical robot setups, which may limit direct comparison with other recent works.

**Reviewer Scores:**

Reviewer EYxJ (Initial Rating: 4, Confidence: 3) : This reviewer likely would have maintained their score or moved to a 5. While the authors clarified the definition of "unified" models and provided latency data , the reviewer's core demand for evaluation on public benchmarks like RoboBrain 2.0 was not met.


Reviewer uSyR (Initial Rating: 6, Confidence: 4) : This reviewer explicitly chose to maintain their score of 6 after the rebuttal. Although they acknowledged that the authors addressed their concerns regarding the synthetic dataset and parallel architectures , they still felt the core impact of the unified structure remained somewhat limited.


Reviewer yjpV (Initial Rating: 6, Confidence: 5) : This reviewer likely would have increased their score to a 7 or became a "strong" 6. The authors provided extensive new quantitative ablation studies specifically requested by this reviewer , demonstrating the necessity of each reasoning component for task success and error recovery.


Reviewer 2sCT (Initial Rating: 6, Confidence: 4) : This reviewer likely would have maintained a 6 or moved to a 7. The authors provided a clear explanation of the supervision scheme for decision tokens , corrected figure ambiguities , and pointed to existing quantitative evidence for error detection that the reviewer had initially overlooked.

---

### Decision · Program_Chairs · 2026-01-26

Accept (Poster)